# General Skeleton Semantics Learning with Probabilistic Masked Context Reconstruction for Skeleton-Based Person Re-Identification

## Abstract

Person re-identification (re-ID) via skeleton data is an emerging topic with immense potential for safety-critical applications. Existing methods usually utilize spatial or temporal skeleton semantics learning (SSL) tasks to facilitate skeleton representation learning, while most SSL tasks are *model-dependent* and lack the ability to capture general fine-grained (*e.g.*, joint-level) spatial-temporal skeleton patterns under different model architectures. To delve into multi-faceted generality of SSL tasks, we first propose an SSL generality assessment framework termed **SCUT** that identifies four key SSL properties: **S**patial-temporal effectiveness, **C**o-training compatibility, **U**nsupervised trainability, and **T**ask transformability. By formulating systematic evaluation criteria for each property, SCUT enables both qualitative and quantitative analysis of SSL generality under varying models and scenarios. Motivated by SCUT to fully harness skeleton context for semantics learning, we further devise a generic ***P**robabilistic Masked Spatial-**T**emporal context **R**econstruction* (**Prompter**) task to enhance performance of skeleton-based person re-ID models. Specifically, Prompter first probabilistically and independently masks joints' structural locations to generate *spatial context*, and then randomly conceal their motion trajectories to form *temporal context*. Through combining both spatial and temporal skeleton context representations to jointly reconstruct and infer skeleton sequences, Prompter encourages the model to capture general valuable spatial-temporal skeleton patterns for person re-ID. Empirical evaluations on SCUT and five benchmark datasets demonstrate the superiority of Prompter to most state-of-the-art SSL tasks. We further validate its general effectiveness in different skeleton modeling, RGB-estimated or cross-domain scenarios[1].

## 1 Introduction

Person re-identification (re-ID) is a vital pattern recognition task to match and retrieve a certain pedestrian from different views or scenes, which has driven many safety-critical applications such as security authentication, smart surveillance, and human tracking (Vezzani et al., 2013; Ye et al., 2021). With recently more accessible skeleton data from low-cost and contactless depth sensors like Kinect (Shotton et al., 2011), skeleton-based person re-ID is attracting increasing attention in both academia and industry (Liao et al., 2020; Rao et al., 2021b; Rao & Miao, 2023). Different from conventional person re-ID methods that require visual appearance or facial features (Wang et al., 2016), skeleton-based methods typically model body structural features and unique motion semantics (*e.g.*, gait (Murray et al., 1964)) from positions of key body joints to identify different persons, which enjoy numerous advantages such as smaller data input, better privacy protection (*e.g.*, without using appearances), and higher robustness to view and background variations (Han et al., 2017).

Early skeleton-based methods (Andersson & Araujo, 2015) extract hand-crafted descriptors such as pairwise joint distances to depict anthropometric and gait attributes of body for person re-ID. However, these methods heavily rely on prior domain knowledge such as kinematics (Yoo et al., 2002) to model skeleton data, which typically lack the ability to fully exploit latent skeleton semantics or features beyond human cognition. To tackle this problem, recent mainstream methods (Liao et al., 2020; Rao

---

[1]Our anonymized codes and models (github.com/Anonymous-9273/Prompter) are publicly available.

& Miao, 2023) resort to deep neural networks to perform skeleton representation learning. In these methods, skeleton semantics learning (SSL) tasks such as skeleton reconstruction are widely adopted as either a main objective (Rao et al., 2020; Rao et al., 2021b;a) or enhancement task (Rao et al., 2021c; Rao & Miao, 2022; Rao & Miao, 2023) to help capture spatial-temporal skeleton patterns and high-level semantics (*e.g.*, skeleton pattern consistency) for person re-ID. Despite the success of existing SSL tasks, most of them are designed based on *certain* model architectures or feature representations (*e.g.*, sequence representations), and often lack the ability to fully mine fine-grained (*e.g.*, joint-level) spatial and temporal semantics for person re-ID. These properties inherently limit their effectiveness and adaptability to different models. On the other hand, current skeleton-based person re-ID endeavors only provide a single performance metric (*i.e.*, accuracy) of SSL tasks under a particular model, but rarely explore and compare different properties associated with their generality under *varying* models or scenarios. For example, they usually lack a general framework to define and quantify key SSL attributes, such as compatibility and trainability (*e.g.*, learnable under unsupervised scenarios), which hinders a fair and multi-faceted evaluation of SSL generality in practice.

To address the above challenges, we first present a systematic generality assessment framework termed **SCUT** that identifies and quantifies key characteristics of SSL tasks in terms of *Spatial-temporal effectiveness (STE)*, *Co-training compatibility (CTC)*, *Unsupervised trainability (UT)*, and *Task transformability (TT)*. Then, based on the SCUT framework, we *for the first time* evaluate the co-training performance of existing state-of-the-art SSL tasks under different models and scenarios (*e.g.*, datasets), and quantitatively measure their general effectiveness in combining spatial and temporal skeleton semantics learning for person re-ID. Our study empirically reveals that existing SSL tasks often exhibit different compatibility (CTC) when applied to other models, and the SSL task that explicitly incorporates spatial-temporal semantics learning (STE) and jointly optimizes multiple sub-tasks (TT) achieves higher performance in most cases. Motivated by these key properties to fully exploit valuable skeletal context information (*e.g.*, structural context of body) for both spatial and temporal pattern learning, we propose a generic *Probabilistic Masked Spatial-Temporal Conext Reconstruction* (**Prompter**) task to enhance general skeleton semantics learning of different models for person re-ID. In particular, Prompter leverages *probabilistic spatial context masking (PSCM)* to probabilistically and independently mask skeletal structural locations (defined as "*spatial context*"), and combines *probabilistic temporal context masking (PTCM)* to generate random partial skeletal motion trajectories (defined as "*temporal context*") to perform complete skeleton sequence reconstruction. Based on the masked spatial and temporal skeleton context representations, Prompter simultaneously *reconstructs* the unmasked positions and *infers* the masked parts of skeleton sequences, so as to encourage the model to capture useful key spatial-temporal skeleton semantics (*e.g.*, structural joint relations) for person re-ID.

The main contributions can be summarized as:

- We identify the key properties of general skeleton semantics learning (SSL) to formulate the *first* SSL generality assessment framework SCUT, and conduct a multi-faceted performance evaluation of of existing state-of-the-art SSL tasks under varying models and scenarios.

- We propose average co-training performance gain and spatial-temporal performance gain to *quantitatively* compare model compatibility (CTC) and spatial-temporal effectiveness (STE) of SSL tasks. We empirically reveal the importance of transformability (TT) in SSL.

- We present a generic SSL task *Prompter* with probabilistic spatial (PSCM) and temporal skeleton context masking (PTCM) for reconstruction and inference of skeleton sequences, which enhances the general spatial-temporal skeleton semantics learning for person re-ID.

- Empirical evaluations on SCUT and five public datasets demonstrate the generality and superiority of Prompter in improving various models, and it is scalable to be applied to RGB-estimated skeletons, cross-domain person re-ID, and different skeleton modeling.

## 2 RELATED WORKS

**Skeleton-Based Person Re-Identification.** Early works extract hand-crafted skeleton descriptors such as anthropometric and gait attributes from body joints for person re-ID: Seven Euclidean distances between certain joint pairs are calculated as discriminative features (Barbosa et al., 2012), while Munaro et al. (2014a) and Pala et al. (2019) further extend it to into 13 ($D_{13}$) and 16 skeleton

descriptors ($D_{16}$) respectively for person re-ID. Recent mainstream methods leverage deep neural networks to learn representations from skeleton sequences or graphs: Liao et al. (2020) propose CNN-based PoseGait to encode joint-based motion descriptors (denoted as $D_{\text{PG}}$) for human recognition. Rao et al. (2020) devise an attention-based encoder-decoder model (AGE) to encode gait features from 3D skeletons, while SGELA (Rao et al., 2021b) further combines sequence contrastive learning to enhance motion semantics learning. A masked contrastive learning framework SimMC is proposed by Rao & Miao (2022) to learn skeleton prototypes and intra-sequence relations for person re-ID. The multi-scale skeleton graphs are explored in MG-SCR (Rao et al., 2021c) and SM-SGE (Rao et al., 2021a) to learn unique body relations and patterns at various levels. TranSG fuses both skeleton-level and sequence-level graph representations for contrastive learning (Rao & Miao, 2023). Some recent person re-ID works also combine RGB images and skeleton data to learn auxiliary anthropometric attributes (Wang et al., 2020), body parts correlations (Lu et al., 2023), and clothing-invariant features (Nguyen et al., 2024) to enhance their performance.

**Skeleton Semantics Learning (SSL).** Learning general spatial-temporal skeleton semantics is pivotal to skeleton-based person re-ID Rao & Miao (2024). The attention-based reconstruction (AR) (Rao et al., 2020) and attention-based contrastive learning (AC) (Rao et al., 2021b) are devised to learn semantics of motion continuity within skeletons. In (Rao et al., 2021c), multi-level skeleton sequence prediction (MSSP) task is proposed based on multi-level graphs, while Rao et al. (2021a) further devise multi-scale skeleton reconstruction (MSR) to capture skeleton dynamics and cross-scale component correspondence. The masked intra-sequence contrastive learning (MIC) is devised in (Rao & Miao, 2022) to learn pattern invariance between different skeleton subsequences. A structure-trajectory prompted reconstruction (STPR) task is proposed in (Rao & Miao, 2023) to learn structural relations and pattern continuity of joints. As far as we know, our work is the first exploration and assessment of multi-faceted generality of existing SSL tasks under different scenarios. Different from previous tasks that rely on certain architectures or sequence-level representations, our method can be generally applied to different models for both spatial and temporal joint-level semantics learning.

## 3 METHOD

### 3.1 PROBLEM FORMULATION

The input skeleton sequence is represented by $\boldsymbol{S} = (\boldsymbol{s}_1, \cdots, \boldsymbol{s}_f) \in \mathbb{R}^{f \times J \times 3}$, where $f$ is the total number of skeletons in the sequence and $\boldsymbol{s}_i \in \mathbb{R}^{J \times 3}$ denotes the $i^{th}$ skeleton with 3D coordinates of $J$ body joints. Each sequence $\boldsymbol{S}$ belongs to an identity $\text{y} \in \{1, \cdots, I\}$ and $I$ is the number of different identity classes. The training set $\Phi_T = \left\{\boldsymbol{S}_i^T\right\}_{i=1}^{N_1}$, probe set $\Phi_P = \left\{\boldsymbol{S}_i^P\right\}_{i=1}^{N_2}$, and gallery set $\Phi_G = \left\{\boldsymbol{S}_i^G\right\}_{i=1}^{N_3}$ contain $N_1$, $N_2$, and $N_3$ skeleton sequences of different persons collected from different scenes or views. The model is trained to encode skeleton sequences into effective representations, such that the encoded representations (denoted as $\{\boldsymbol{V}_i^P\}_{i=1}^{N_2}$) in the probe set can be matched with the representations (denoted as $\{\boldsymbol{V}_i^G\}_{i=1}^{N_3}$) of the same identity in the gallery set.

The focus of this study is to devise a general SSL task that can be applied to different models (denoted as *base models*) to learn more effective *spatial-temporal* skeleton semantics to improve person re-ID performance. Formally, we denote a base model as $F_\theta(\cdot)$ with the randomly-initialized learnable parameters $\theta$, and the model encoding process of skeleton sequences can be formulated as

$$F_\theta(\boldsymbol{S}) = \boldsymbol{V} = [\boldsymbol{v}_1; \boldsymbol{v}_2; \cdots; \boldsymbol{v}_f], \tag{1}$$

where the optimal parameters $\theta^*$ can be obtained by

$$\theta^* = \arg\min_\theta \left[\lambda \mathcal{L}_{\text{D}} + (1 - \lambda)\mathcal{L}_{\text{SSL}}\right]. \tag{2}$$

In Eq. (1) and (2), $\boldsymbol{v}_t \in \mathbb{R}^{J \times H}$ ($t \in \{1, 2, \cdots, f\}$) represents the the $t^{th}$ skeleton representation concatenated by $J$ encoded body-joint representations with the embedding size $H$, $[; ;]$ denotes the feature concatenation operation, $\theta^*$ represents the optimal model parameters by jointly minimizing downstream task objective loss $\mathcal{L}_{\text{D}}$ (*e.g.*, classification loss) and skeleton semantics learning objective loss $\mathcal{L}_{\text{SSL}}$ (*e.g.*, reconstruction loss), and $\lambda$ is the weight coefficient to fuse different losses. For simplicity, we use $\boldsymbol{S}$ and $\boldsymbol{V}$ to denote the training skeleton sequence $\boldsymbol{S}_i^T$ and its encoded representation $\boldsymbol{V}_i^T$, respectively. It is worth noting that the SSL task ($\mathcal{L}_{\text{SSL}}$) typically plays an equally-important

role as downstream objective (*i.e.*, $\lambda = 0.5$) for skeleton representation learning, and can also serve as a main task (*i.e.*, $\lambda = 0.0$) in self-supervised learning paradigms (Rao et al., 2021a).

## 3.2 GENERALITY ASSESSMENT OF SKELETON SEMANTICS LEARNING

To evaluate multi-faceted generality of an SSL task across different models and scenarios, we propose a generality assessment framework (SCUT) with four key quantitative and qualitative characteristics.

**Quantitative Properties:**

- **Co-Training Compatibility (CTC).** An SSL task with general applicability should be compatible with different architectures and downstream objectives. In particular, if an SSL task can be co-trained with different models, and achieve higher performance than the original models *on average*, this SSL task is eligible to possess CTC. In principle, CTC requires that the SSL task can be performed on the original skeleton representations without necessitating the construction of a new independent architecture or component. Here we define the *average co-training performance gain* to quantify the CTC of an SSL task by

$$G_{\mathrm{C}} = \frac{1}{N_{\mathrm{m}} N_{\mathrm{d}}} \sum_{i=1}^{N_{\mathrm{m}}} \sum_{j=1}^{N_{\mathrm{d}}} \gamma_{i,j} \frac{\overline{A^*}_{i,j} - \overline{A}_{i,j}}{\overline{A}_{i,j}}, \tag{3}$$

  where $G_{\mathrm{C}} \in (-1, 1)$ is the average co-training performance gain under a common assumption that the absolute value of performance change after applying SSL does NOT exceed the original performance value, $\overline{A}_{i,j}$ and $\overline{A^*}_{i,j}$ respectively denote the average performance of the $i^{th}$ applied base model on the $j^{th}$ dataset without SSL and employing the SSL task, $\gamma_{i,j}$ represents the weight coefficient to evaluate the SSL task on the combination of $i^{th}$ model and $j^{th}$ dataset, $N_{\mathrm{m}}$ and $N_{\mathrm{d}}$ represents the number of different applied base models and different datasets respectively. We adopt the most frequently used metric, Rank-1 accuracy, as the performance indicator, and average their results when applied to different base models on varying datasets (see Sec. 3). It is worth noting that we use four most common benchmark datasets to evaluate SLL tasks, and consider each applied base model and dataset equally significant (*i.e.*, $\gamma_{i,j} = 1$). Intuitively, a larger $G_{\mathrm{C}}$ value indicates incorporating the SSL task into different models can achieve higher average accuracy improvement under varying datasets (*i.e.*, data distributions), thereby suggesting its better compatibility and adaptability.

- **Spatial-Temporal Effectiveness (STE).** As the core of skeleton-based person re-ID is to capture both spatial body features and temporal motion patterns to discriminate different persons (Rao & Miao, 2023), an SSL task is considered to possess higher general effectiveness if it *explicitly* contains both spatial and temporal modeling (*e.g.*, body structure and trajectory dynamics) of skeleton data. STE requires that both temporal and spatial part in the SSL task are effective (*i.e.*, each part can individually improve the model performance), and can be compatibly combined to achieve further improvement. The *average spatial-temporal performance gain* is defined to measure the overall STE of an SSL task with

$$G_{\mathrm{ST}} = \frac{1}{N_{\mathrm{m}} N_{\mathrm{d}}} \sum_{i=1}^{N_{\mathrm{m}}} \sum_{j=1}^{N_{\mathrm{d}}} \gamma_{i,j} R_{i,j}^{\mathrm{ST}} \frac{\overline{A^*}_{i,j} - \overline{A}_{i,j}}{\overline{A}_{i,j}}, \tag{4}$$

  where

$$R_{i,j}^{\mathrm{ST}} = \frac{\min(\overline{A^{\mathrm{S}}}_{i,j} - \overline{A}_{i,j}, \overline{A^{\mathrm{T}}}_{i,j} - \overline{A}_{i,j})}{\max(\overline{A^{\mathrm{S}}}_{i,j} - \overline{A}_{i,j}, \overline{A^{\mathrm{T}}}_{i,j} - \overline{A}_{i,j})}. \tag{5}$$

  In Eq. (4) and (5), $G_{\mathrm{ST}} \in (-1, 1)$ denotes the average spatial-temporal performance gain following the same notation of Eq. (3), $\overline{A^{\mathrm{S}}}_{i,j}$ and $\overline{A^{\mathrm{T}}}_{i,j}$ denote the average performance of the $i^{th}$ base model on the $j^{th}$ dataset when applying only the spatial component or temporal component of the SSL task, $\min(a, b)$ and $\max(a, b)$ denote the minimum and maximum value between $a$ and $b$. Here we adopt the *relative* ratio $R_{i,j}^{\mathrm{ST}} \in (0, 1]$ between the performance gains of spatial part and temporal part (see Eq. (5)) as the scale coefficient to consider the balance of spatial and temporal effectiveness: A good STE requires that both temporal and spatial part can equally or similarly contribute to the performance improvement

Table 1: Generality assessment of SSL based on four key properties. "DR" represents direct skeleton reconstruction. ✔ indicates satisfying the corresponding property. "+" denotes combining tasks.

| ID | SSL Task | Quantitative | | Qualitative | | Generality |
| --- | --- | --- | --- | --- | --- | --- |
| | | CTC ($G_C$) (%) | STE ($G_{ST}$) (%) | UT | TT | $\widehat{G}$ |
| 1 | DR | 2.66 | ✗ | ✔ | ✗ | 0.3783 |
| 2 | AR (Rao et al., 2020) | ✗ | ✗ | ✔ | ✗ | 0.2500 |
| 3 | AR + AC (Rao et al., 2021b) | ✗ | ✗ | ✔ | ✗ | 0.2500 |
| 4 | MSSP (Rao et al., 2021c) | ✗ | ✗ | ✔ | ✗ | 0.2500 |
| 5 | MSR (Rao et al., 2021a) | ✗ | ✗ | ✔ | ✔ | 0.5000 |
| 6 | MIC (Rao & Miao, 2022) | 3.86 | ✗ | ✔ | ✗ | 0.3798 |
| 7 | STPR (Rao & Miao, 2023) | 7.19 | 1.62 | ✔ | ✗ | 0.5110 |
| **8** | **Prompter (Ours)** | **9.50** | **4.49** | ✔ | ✔ | **0.7675** |

(*i.e.*, $R_{i,j}^{ST} \to 1$). When a part offers extremely overwhelming performance gain (*i.e.*, $R_{i,j}^{ST} \ll 1$) compared to the other part, it suggests that the other part possibly provides very slight contribution to the improvement. Thus, a large $R_{i,j}^{ST}$ indicates that both parts possess independent effectiveness and their combination is empirically meaningful to improve the performance. $G_{ST}$ incorporates the contribution of both spatial and temporal components of an SSL task to indicate the average performance gain for their spatial-temporal combination.

**Qualitative Attributes:**

- **Unsupervised Trainability (UT).** An SSL task without using class labels can be trained in more general scenarios (*e.g.*, unsupervised skeleton learning). The UT property guarantees that the SSL task can be commonly applied to different datasets and unknown classes (*i.e.*, class-agnostic). In practice, it encourages the model to learn class-agnostic general skeleton semantics (*e.g.*, universal motion patterns), which can be combined with class-specific learning of the downstream task objective to enhance person re-ID performance.

- **Task Transformability (TT).** Transformability is a key attribute in general SSL tasks, as it enables flexibly adapting the semantics learning objective to a certain architecture or downstream task. If an SSL task explicitly contains other SSL tasks (defined as *sub-tasks*) or can be potentially transformed to them under different probabilities, this SSL task is considered to possess transformability (TT). For example, an SSL task that directly combines reconstruction and prediction tasks possesses the TT property. Performing such task can be viewed as to simultaneously optimize different SSL sub-tasks (see Sec. 4.3), therefore often possessing higher generality and effectiveness than same-type SSL tasks without TT.

By synergizing the above key criteria, SCUT computes the final generality score of an SSL task with:

$$\widehat{G} = \omega_1[(\omega_C G_C + (1 - \omega_C))\,\mathbb{I}(\textbf{CTC})] + \omega_2[(\omega_{ST} G_{ST} + (1 - \omega_{ST}))\mathbb{I}(\textbf{STE})] + \omega_3\mathbb{I}(\textbf{UT}) + \omega_4\mathbb{I}(\textbf{TT}), \quad (6)$$

where $0 \le \widehat{G} \le 1$ is the normalized score of generality, $\mathbb{I}(\cdot)$ represents the indicator function with value 1 if the SSL task possesses the corresponding property otherwise value is 0, $\omega_1, \omega_2, \omega_3, \omega_4$ are weight coefficients with $\omega_1 + \omega_2 + \omega_3 + \omega_4 = 1$ to combine scores of different properties, $\omega_C$ and $\omega_{ST}$ are coefficients to integrate the basic score and the average performance gain by CTC (*i.e.*, $G_C$) and STE (*i.e.*, $G_{ST}$). It should be noted that the score for average performance gain is added only when the SSL task possesses corresponding property (*i.e.*, $\mathbb{I}(\cdot) = 1$). As we equally focus on each property and their achieved average performance gain in measuring the overall generality of SSL, we assign the same weight value to each of their scores in Eq. (6).

**Generality Comparison of State-of-the-Art SSL Tasks.** As shown in Table 1, unsupervised trainability (UT) is the most common attribute of SSL tasks, as existing SSL tasks are typically designed for unlabeled skeleton learning and can learn effective general class-agnostic semantics. However, only four tasks (ID = 1, 6, 7, 8) can be flexibly applied to different models without constructing new model architectures or components, while the proposed Prompter (ID = 8) presents higher co-training compatibility than other three tasks with a significant improvement of $2.31\%$ to $6.84\%$ average performance gain on varying models and datasets (shown in Table 2). For SSL tasks that explicitly model spatial and temporal skeleton patterns, our method also shows the strongest spatial-temporal effectiveness (STE), achieving more than twice the performance gain of the state-of-the-art SSL task STPR (ID = 7). This demonstrates that Prompter could possess more balanced

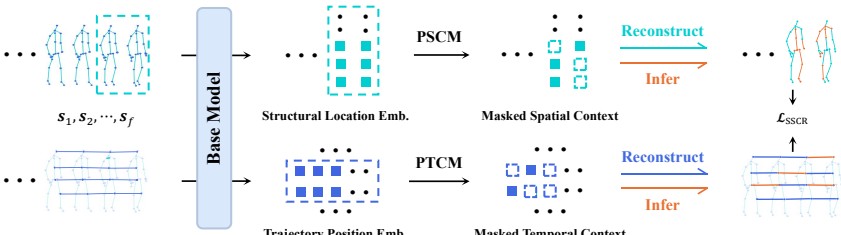

Figure 1: Schematics of Prompter: First, the structural locations and motion trajectories of body joints in the input skeleton sequence $s_1, \cdots, s_f$ are encoded by the base model into feature embedding (Emb.). Then, we apply PSCM and PTCM to probabilistically and independently mask the location and trajectory embeddings to generate random spatial and temporal skeleton context representations, which are exploited to reconstruct and infer the complete skeletal locations and trajectory by minimizing $\mathcal{L}_{\text{SSCR}}$ to learn general and valuable spatial-temporal semantics for person re-ID.

effective spatial and temporal semantics learning with higher combined performance. Interestingly, only MSR (ID = 5) and Prompter (ID = 8) possess task transformability (TT): MSR contains DR and cross-scale skeleton inference, while the proposed Prompter can be viewed as to perform DR and STPR under different probabilities (see Sec. 3.3). Notably, our method (ID = 8) simultaneously satisfies all key properties and achieves the highest generality score, suggesting that it could be more flexibly and effectively applied to different models and scenarios. We further show the importance and key effects of different properties such as TT and STE in SSL (analyzed in Sec. 3.3, 4.2, and 4.3).

### 3.3 PROBABILISTIC MASKED SPATIAL-TEMPORAL CONTEXT RECONSTRUCTION

To realize general and effective SSL, it is essential to align its objective to the key of skeleton-based person re-ID, which aims to capture both spatial skeleton features (*e.g.*, body structural features) and temporal motion attributes (*e.g.*, gait (Murray et al., 1964)). Such spatial and temporal patterns can be respectively characterized by positions and relations of different body joints within each skeleton and their corresponding trajectories. A straightforward method is to perform skeleton reconstruction without explicit temporal or spatial modeling (Rao et al., 2021b), while such task usually lacks the flexibility to fully exploit varying valuable context information (*e.g.*, temporal context of trajectory) of fine-grained skeleton representations (*e.g.*, body joints) to capture richer skeleton semantics. According to the crucial properties of SSL identified by SCUT (see Sec. 3.2), a more general solution to these challenges is to explicitly model both spatial and temporal skeleton patterns (*i.e.*, implement STE) while combining multiple skeleton context based learning sub-objectives (*i.e.*, establish TT) to build a more effective SSL task. To this end, we propose ***Pr**obabilistic **M**asked **S**patial-**T**emporal Cont**e**xt **R**econstruction* (**Prompter**) that randomly and independently masks structural locations of joints (defined as "*skeletal spatial context*") and motion trajectories of joints (defined as "*skeletal temporal context*") to jointly reconstruct and infer spatial-temporal context (*e.g.*, masked positions) of skeleton sequences, so as to learn general effective skeleton semantics for person re-ID.

**Probabilistic Spatial Context Masking (PSCM).** Given the $t^{th}$ skeleton representation $\boldsymbol{v}_t$ with structural locations of $J$ body joints, $\boldsymbol{v}_t^1, \boldsymbol{v}_t^2, \cdots, \boldsymbol{v}_t^J$, we mask their spatial context by randomly discarding each location with a probability $p_{\text{S}}$. The spatially-masked skeleton representation is obtained by

$$\widetilde{\boldsymbol{v}}_t = \frac{1}{n_{\text{S}}} \sum_{j=1}^{J} x_j \boldsymbol{v}_t^j, \tag{7}$$

where $\widetilde{\boldsymbol{v}}_t \in \mathbb{R}^H$ denotes the masked spatial context representation of $t^{th}$ skeleton after applying PSCM, $x_j \in \{0, 1\}$ is the $j^{th}$ location mask constructed by an *independent and identically distributed (IID)* Bernoulli random variable with the probability $p_{\text{S}}$ of being 0 (*i.e.*, $x_j \sim \text{Bernoulli}(1 - p_{\text{S}})$), $n_{\text{S}} = \sum_{j=1}^{J} x_j$ denotes the number of unmasked structural locations, and $n_{\text{S}} \geq 1$ is used to avoid empty skeleton context. Each body-joint location in the skeleton is assumed to be equally important and we average them to generate the spatial context representation. It is noteworthy that PSCM can be extended with other probabilistic distributions, and we adopt the commonly-used Bernoulli distribution due to its simplicity and computational tractability (Boluki et al., 2020).

**Probabilistic Temporal Context Masking (PTCM).** Provided the motion trajectory (*i.e.*, temporal positions), $\boldsymbol{v}_1^i, \boldsymbol{v}_2^i, \cdots, \boldsymbol{v}_f^i$, of the $i^{th}$ body joint, we mask their temporal context by randomly dropping each trajectory position with a probability $p_T$ as (shown in Fig. 1)

$$\overline{\boldsymbol{w}}^i = \frac{1}{n_T} \sum_{t=1}^{f} z_t \boldsymbol{v}_t^i. \tag{8}$$

In Eq. (8), $\overline{\boldsymbol{w}}^i \in \mathbb{R}^H$ denotes the masked temporal context representation of $i^{th}$ joint after applying PTCM, and the mask $z_t \in \{0, 1\}$ is an IID Bernoulli random variable with $z_t \sim \text{Bernoulli}(1 - p_T)$, $n_T = \sum_{t=1}^{f} z_T$ represents the number of unmasked positions in the $i$-joint motion trajectory, and $n_T \geq 1$ is adopted to generate non-empty temporal context. Each position in the motion trajectory is assigned with equal importance and we average them to obtain the temporal context representation.

**Skeleton Context Reconstruction Objective.** Based on the masked spatial and temporal context representation, $\widetilde{\boldsymbol{v}}$ (see Eq. (7)) and $\overline{\boldsymbol{w}}$ (See Eq. (8)), we propose the Spatial-temporal Skeleton Context Reconstruction (SSCR) loss to reconstruct and infer the original skeleton sequences with

$$\mathcal{L}_{\text{SSCR}} = \frac{\alpha}{N_1} \sum_{i=1}^{N_1} \left\| \widetilde{\boldsymbol{S}}_i - \boldsymbol{S}_i \right\|_2^2 + \frac{(1-\alpha)}{N_1} \sum_{i=1}^{N_1} \left\| \overline{\boldsymbol{S}}_i - \boldsymbol{S}_i \right\|_2^2, \tag{9}$$

where the spatially and temporally predicted skeleton sequences are respectively represented by

$$\widetilde{\boldsymbol{S}}_i = \left[ \left\|_{t=1}^{f} \Phi_S(\mathcal{R}_S(\widetilde{\boldsymbol{v}}_t); \mathcal{I}_S(\widetilde{\boldsymbol{v}}_t)) \right], \tag{10}$$

$$\overline{\boldsymbol{S}}_i = \left[ \left\|_{j=1}^{J} \Phi_T(\mathcal{R}_T(\overline{\boldsymbol{w}}^j); \mathcal{I}_T(\overline{\boldsymbol{w}}^j)) \right]^\top. \tag{11}$$

In Eq. (9), $\alpha$ is the weight coefficient to combine spatial and temporal skeleton context reconstruction, $\boldsymbol{S}_i, \widetilde{\boldsymbol{S}}_i, \overline{\boldsymbol{S}}_i \in \mathbb{R}^{f \times J \times 3}$ denote the $i^{th}$ ground-truth training skeleton sequence, the $i^{th}$ predicted skeleton sequences using spatial and temporal masked context representations respectively, and $\| \cdot \|_2$ represents the $\ell_2$ norm. During the context reconstruction process (see Eq. (10) and (11)), the objective of $\mathcal{L}_{\text{SSCR}}$ not only **reconstructs** the structural locations and trajectory positions that correspond to the *unmasked* skeleton context using reconstructing models $\mathcal{R}_S(\cdot)$ and $\mathcal{R}_T(\cdot)$, but also **infers** the *masked* spatial and temporal positions based on the partial context (*i.e.*, unmasked context representations) using inferring models $\mathcal{I}_S(\cdot)$ and $\mathcal{I}_T(\cdot)$. Both reconstructing and inferring models adopt identical architectures built by multi-layer perceptron (MLP) networks. $\Phi_S(\cdot)$ and $\Phi_T(\cdot)$ denote permutation functions to sort predicted joint positions in a default spatial and temporal order based on the pre-defined indices. $\|$ denotes concatenating $f$ skeletons or $J$ body-joint trajectories to form the skeleton sequence. For convenience, we use $^\top$ to denote transposing the trajectory position matrix from $\mathbb{R}^{J \times f \times 3}$ to $\mathbb{R}^{f \times J \times 3}$ to match the shape of original skeleton sequences.

By employing $\mathcal{L}_{\text{SSCR}}$, Prompter essentially exploits incomplete structural and motion information as partial context to *prompt* the model for complete context reconstruction with an inference of *unknown* spatial and temporal positions. This inherently requires the model to effectively comprehend and utilize useful spatial (*e.g.*, key structural relations of joints) and temporal skeleton semantics (*e.g.*, motion continuity) to achieve precise reconstruction and inference, which facilitates the model to capture more valuable spatial-temporal skeleton features for person re-ID.

**Generalization of Prompter.** The Prompter task can be viewed as a general probabilistic form of existing reconstruction or masked reconstruction based SSL tasks (Rao et al., 2021a; Rao & Miao, 2023). It owns the task transformability (TT): The direct spatial reconstruction with all body joints unmasked is contained in Prompter with the probability of $\mathcal{P}_S(J) = (1 - p_S)^J$ (see Appendix II), while the masked spatial skeleton reconstruction with $n_S$ unmasked joints is a special case of Prompter with the occurrence probability of $\mathcal{P}_S(n_S) = \binom{J}{n_S}(p_S)^{J-n_S}(1 - p_S)^{n_S}$. This enables it to jointly optimize different SSL sub-tasks and achieve better semantics learning performance (see Sec. 4.2 and 4.3). Intuitively, Prompter introduces more possible spatial-temporal reconstruction cases (*i.e.*, under varying partial spatial and temporal contexts) than both direct reconstruction and masked reconstruction (Rao & Miao, 2023) that employs a *fixed* number of masks, thereby potentially improving the reconstruction diversity to reduce model over-fitting. We further reveal its relations to model regularization methods (*e.g.*, Dropout (Baldi & Sadowski, 2014)) in the appendices.

Table 2: Performance evaluation of our method when applied to four state-of-the-art methods on different datasets. We also include representative hand-crafted, supervised (♠), self-supervised and unsupervised (♢) methods as performance reference. "+" denotes employing Prompter to co-train models. The **bold numbers** indicate higher performance than the base model *without* using SSL.

| Methods | KS20 | | | | BIWI-W | | | | BIWI-S | | | | IAS-A | | | | IAS-B | | | | KGBD | | | |
|---|---|---|---|---|---|---|---|---|---|---|---|---|---|---|---|---|---|---|---|---|---|---|---|---|
| | mAP | $R_1$ | $R_5$ | $R_{10}$ | mAP | $R_1$ | $R_5$ | $R_{10}$ | mAP | $R_1$ | $R_5$ | $R_{10}$ | mAP | $R_1$ | $R_5$ | $R_{10}$ | mAP | $R_1$ | $R_5$ | $R_{10}$ | mAP | $R_1$ | $R_5$ | $R_{10}$ |
| $D_{PG}$ (Liao et al., 2020) | 11.3 | 35.2 | 61.5 | 70.5 | 8.7 | 6.5 | 15.5 | 20.3 | 6.7 | 18.5 | 45.4 | 63.8 | 11.0 | 16.4 | 39.5 | 53.4 | 10.6 | 16.0 | 41.2 | 57.3 | 2.1 | 30.0 | 49.1 | 58.1 |
| $D_{13}$ (Munaro et al., 2014a) | 18.9 | 39.4 | 71.7 | 81.7 | 17.2 | 14.2 | 20.6 | 23.7 | 13.1 | 28.3 | 53.1 | 65.9 | 24.5 | 40.0 | 58.7 | 67.6 | 23.7 | 43.7 | 68.6 | 76.7 | 1.9 | 17.0 | 34.4 | 44.2 |
| $D_{16}$ (Pala et al., 2019) | 24.0 | 51.7 | 77.1 | 86.9 | 18.8 | 17.0 | 25.3 | 29.6 | 16.7 | 32.6 | 55.7 | 68.3 | 25.2 | 42.7 | 62.9 | 70.7 | 24.5 | 44.5 | 69.1 | 80.2 | 4.0 | 31.2 | 50.9 | 59.8 |
| PoseGait♠ (Liao et al., 2020) | 23.5 | 49.4 | 80.9 | 90.2 | 11.1 | 8.8 | 23.0 | 31.2 | 9.9 | 14.0 | 40.7 | 56.7 | 17.5 | 28.4 | 55.7 | 69.2 | 20.8 | 28.9 | 51.6 | 62.9 | 13.9 | 50.6 | 67.0 | 72.6 |
| AGE♢ (Rao et al., 2020) | 8.9 | 43.2 | 70.1 | 80.0 | 12.6 | 11.7 | 21.4 | 27.3 | 8.9 | 25.1 | 43.1 | 61.6 | 13.4 | 31.1 | 54.8 | 67.4 | 12.8 | 31.1 | 52.3 | 64.2 | 0.9 | 2.9 | 5.6 | 7.5 |
| SGELA♢ (Rao et al., 2021b) | 21.2 | 45.0 | 65.0 | 75.1 | 19.0 | 11.7 | 14.0 | 14.7 | 15.1 | 25.8 | 51.8 | 64.4 | 13.2 | 16.7 | 30.2 | 44.0 | 14.0 | 22.2 | 40.8 | 50.2 | 4.5 | 38.1 | 53.5 | 60.0 |
| SM-SGE♢ (Rao et al., 2021a) | 9.5 | 45.9 | 71.9 | 81.2 | 15.2 | 13.2 | 25.8 | 33.5 | 10.1 | 31.3 | 56.3 | 69.1 | 13.6 | 34.0 | 60.5 | 71.6 | 13.3 | 38.9 | 64.1 | 75.8 | 4.4 | 38.2 | 54.2 | 60.7 |
| MG-SCR♠ (Rao et al., 2021c) | 11.3 | 49.0 | 69.3 | 80.3 | 12.7 | 35.6 | 60.7 | 72.2 | 12.6 | 34.2 | 60.4 | 72.5 | 17.1 | 45.6 | 70.0 | 80.3 | 18.5 | 49.7 | 72.3 | 82.1 | 5.5 | 48.2 | 66.4 | 72.5 |
| **+ Promter (Ours)** | **13.2** | **56.3** | **76.0** | **82.4** | **13.5** | **39.5** | **65.1** | **75.6** | **13.4** | **37.6** | **64.5** | **75.9** | **20.1** | **52.7** | **74.4** | **82.8** | **20.5** | **51.7** | **73.8** | **83.4** | **7.0** | **50.9** | **67.3** | **73.0** |
| SPC-MGR♢ (Rao & Miao, 2022) | 21.7 | 59.0 | 79.0 | 86.2 | 19.4 | 18.9 | 31.5 | 40.5 | 16.0 | 34.1 | 57.3 | 69.8 | 24.2 | 41.9 | 66.3 | 75.6 | 24.1 | 43.3 | 68.4 | 79.4 | 6.9 | 40.8 | 57.5 | 65.0 |
| **+ Promter (Ours)** | **23.7** | **65.0** | **79.8** | **85.7** | 18.9 | **37.0** | **61.6** | **74.5** | 18.9 | **37.6** | **67.2** | **78.8** | **27.1** | **49.8** | **73.1** | **80.8** | **28.1** | **51.0** | **73.4** | **83.0** | **7.7** | **41.5** | **58.3** | **65.4** |
| SimMC♢ (Rao & Miao, 2022) | 21.1 | 65.6 | 81.0 | 86.9 | 19.5 | 23.7 | 36.4 | 44.2 | 11.7 | 40.1 | 63.2 | 74.2 | 18.5 | 43.1 | 65.1 | 72.3 | 22.3 | 43.8 | 67.0 | 74.9 | 11.0 | 53.6 | 65.2 | 70.5 |
| **+ Promter (Ours)** | **22.3** | **67.8** | **82.3** | **87.5** | **20.0** | **24.5** | **37.2** | **44.9** | **12.3** | **42.8** | **65.8** | **75.6** | **21.5** | **46.0** | **66.2** | **75.1** | **24.0** | **47.0** | 66.9 | **76.0** | **12.0** | **55.2** | **66.6** | **71.3** |
| TranSG♠ (Rao & Miao, 2023) | 42.5 | 71.3 | 85.4 | 88.9 | 25.5 | 31.2 | 44.9 | 50.7 | 26.7 | 66.6 | 83.6 | 91.4 | 31.8 | 48.0 | 65.5 | 71.8 | 37.9 | 56.1 | 77.5 | 85.1 | 18.1 | 57.0 | 68.0 | 73.4 |
| **+ Promter (Ours)** | **48.3** | **74.2** | **88.0** | **90.7** | **27.3** | **34.6** | **60.9** | **70.2** | **30.3** | **66.8** | **87.3** | **92.2** | **34.1** | **49.5** | **67.8** | **74.3** | **43.8** | **60.4** | **77.9** | **86.5** | **21.3** | **59.5** | **73.0** | **78.3** |

Table 3: Performance comparison of different SSL tasks when applied to state-of-the-art models on different datasets. "+" denotes using the corresponding SSL task. ‡ indicates the model without using any SSL task, and ∗ refers to the original task used in the model. The amount of network parameters (million (M)) and computational complexity (giga foating-point operations (GFLOPs)) for the base model employing a different SSL task are reported. **Bold numbers** denote the best performance among compared SSL tasks, while the underline represents the best results among all methods.

| Methods | Params | GFLOPs | KS20 | | | | BIWI-W | | | | BIWI-S | | | | IAS-A | | | | IAS-B | | | | KGBD | | | |
|---|---|---|---|---|---|---|---|---|---|---|---|---|---|---|---|---|---|---|---|---|---|---|---|---|---|---|
| | | | mAP | $R_1$ | $R_5$ | $R_{10}$ | mAP | $R_1$ | $R_5$ | $R_{10}$ | mAP | $R_1$ | $R_5$ | $R_{10}$ | mAP | $R_1$ | $R_5$ | $R_{10}$ | mAP | $R_1$ | $R_5$ | $R_{10}$ | mAP | $R_1$ | $R_5$ | $R_{10}$ |
| ‡ MG-SCR♠ | | | 11.3 | 49.0 | 69.3 | 80.3 | 12.7 | 35.6 | 60.7 | 72.2 | 12.6 | 34.2 | 60.4 | 72.5 | 17.1 | 45.6 | 70.0 | 80.3 | 18.5 | 49.7 | 72.3 | 82.1 | 5.5 | 48.2 | 66.4 | 72.5 |
| + MIC | 0.37 | 7.39 | **15.9** | 54.1 | 75.2 | 81.5 | 12.0 | 29.9 | 54.8 | 69.1 | **13.8** | 37.0 | 62.3 | 74.2 | 18.4 | 46.6 | 66.6 | 75.0 | 17.8 | 49.0 | 72.2 | 81.0 | 5.3 | 46.5 | 63.6 | 70.4 |
| + DR | 0.99 | 8.44 | 13.1 | 52.0 | 72.1 | 80.5 | 13.1 | 34.1 | 63.2 | 74.9 | 13.0 | 35.0 | 62.8 | 74.5 | 18.3 | 46.6 | 73.1 | 82.6 | 16.4 | 45.6 | 71.0 | 80.5 | 4.9 | 46.8 | 64.8 | 71.8 |
| + STPR | 0.35 | 7.31 | 13.0 | 53.3 | 74.0 | 82.4 | 12.7 | 36.3 | 62.9 | 74.9 | 12.9 | 36.8 | 61.6 | 72.8 | 18.4 | 52.3 | 73.4 | 82.3 | 19.9 | 51.4 | 73.2 | 81.7 | 5.6 | 48.7 | 65.2 | 71.8 |
| + Promter (Ours) | 0.35 | 7.31 | 13.2 | **56.3** | **76.0** | **82.4** | **13.5** | **39.5** | **65.1** | **75.6** | 13.4 | 37.6 | **64.5** | **75.9** | **20.1** | **52.7** | **74.4** | **82.8** | **20.5** | **51.7** | **73.8** | **83.4** | **7.0** | **50.9** | **67.3** | **73.0** |
| ‡ SPC-MGR♢ | | | 21.7 | 59.0 | 79.0 | 86.2 | 19.4 | 18.9 | 31.5 | 40.5 | 16.0 | 34.1 | 57.3 | 69.8 | 24.2 | 41.9 | 66.3 | 75.6 | 24.1 | 43.3 | 68.4 | 79.4 | 6.9 | 40.8 | 57.5 | 65.0 |
| + MIC | 0.01 | 0.75 | 23.4 | 62.9 | 79.0 | 84.0 | 14.8 | 35.1 | **62.3** | **75.1** | 13.5 | 36.3 | 65.0 | 75.1 | 22.9 | 47.0 | 70.0 | 79.3 | 26.9 | 49.3 | 71.8 | 82.8 | 7.5 | 40.6 | 57.9 | 65.0 |
| + DR | 0.22 | 1.38 | 23.2 | 64.2 | 78.7 | 83.2 | 14.3 | 33.8 | 62.1 | 72.7 | 14.6 | 37.1 | 66.8 | 78.3 | 26.6 | 48.7 | 72.8 | **82.1** | 23.4 | 45.9 | 70.7 | 79.9 | 6.6 | 39.2 | 53.3 | 61.4 |
| + STPR | 0.01 | 0.71 | 23.3 | 64.5 | 79.3 | **85.7** | 14.5 | 35.9 | 62.1 | 74.2 | 12.8 | 36.7 | 64.5 | 77.4 | 23.9 | 45.2 | 68.6 | 77.2 | 23.4 | 49.6 | 71.3 | 82.7 | 7.0 | 41.5 | 56.0 | 63.8 |
| + Promter (Ours) | 0.01 | 0.71 | **23.7** | **65.0** | **79.8** | **85.7** | **18.9** | **37.0** | 61.6 | 74.5 | **15.0** | **37.7** | **67.2** | **78.8** | **27.1** | **49.8** | **73.1** | 80.9 | **28.1** | **51.0** | **73.4** | **83.0** | **7.7** | **41.9** | **58.3** | **65.4** |
| ‡ SimMC♢ | | | 21.1 | 65.6 | 81.0 | 86.9 | 19.5 | 23.7 | 36.4 | 44.2 | 11.7 | 40.1 | 63.2 | 74.2 | 18.5 | 43.1 | 65.1 | 72.3 | 22.3 | 43.8 | 67.0 | 74.9 | 11.0 | 53.6 | 65.2 | 70.5 |
| + MIC∗ | 0.15 | 0.95 | **22.3** | 66.4 | 80.7 | 87.0 | 19.9 | **24.5** | 36.7 | 44.5 | **12.3** | 41.7 | **66.6** | **76.8** | 18.7 | 44.8 | 65.3 | 72.9 | 22.9 | 46.3 | **68.1** | **77.0** | 11.7 | 54.9 | 66.2 | 70.6 |
| + DR | 3.06 | 9.00 | 20.1 | 64.5 | 79.3 | 85.2 | 19.4 | 24.1 | 35.0 | 43.0 | 10.9 | 41.0 | 66.0 | 75.0 | 19.0 | 40.4 | 61.2 | 69.3 | 21.4 | 42.6 | 63.1 | 72.9 | 10.4 | 53.8 | 64.5 | 69.5 |
| + STPR | 1.57 | 5.36 | 21.0 | 66.9 | 80.7 | 87.1 | 19.8 | 24.4 | 36.7 | 43.4 | 11.9 | 42.1 | 66.4 | 75.1 | 19.5 | 45.0 | 64.0 | 72.4 | 23.6 | 46.7 | 65.3 | 74.2 | 11.7 | 55.4 | 66.3 | 71.0 |
| + Promter (Ours) | 1.57 | 5.36 | **22.3** | **67.8** | **82.3** | **87.5** | **20.0** | **24.5** | **37.2** | **44.9** | **12.3** | **42.8** | 65.8 | 75.6 | **21.5** | **46.0** | **66.2** | **75.1** | **24.0** | **47.0** | 66.9 | 76.0 | **12.0** | **55.5** | **66.6** | **71.3** |
| ‡ TranSG♠ | | | 42.5 | 71.3 | 85.4 | 88.9 | 25.5 | 31.2 | 44.9 | 50.7 | 26.7 | 66.6 | 83.6 | 91.4 | 31.8 | 48.0 | 65.5 | 71.8 | 37.9 | 56.1 | 77.5 | 85.1 | 18.1 | 57.0 | 68.0 | 73.4 |
| + MIC | 0.42 | 33.75 | 47.2 | 72.3 | 86.1 | 90.2 | 17.7 | 34.5 | 59.8 | 68.8 | 31.5 | 60.0 | 83.0 | 88.3 | 33.0 | 45.2 | 63.5 | 70.7 | 41.8 | 59.4 | 75.7 | 83.0 | 12.1 | 52.1 | 66.6 | 72.3 |
| + DR | 0.41 | 33.69 | 47.8 | 73.2 | 86.7 | 90.4 | 22.0 | 33.8 | 56.5 | 68.9 | 29.0 | 63.5 | 85.6 | 92.0 | 32.1 | 48.2 | 66.2 | 71.6 | 42.8 | 58.0 | 75.2 | 81.6 | 13.3 | 52.8 | 66.6 | 72.9 |
| + STPR∗ | 0.40 | 20.19 | 46.2 | 73.6 | 86.3 | 90.2 | 26.9 | 32.7 | 44.9 | 52.2 | 30.1 | 68.7 | 86.5 | 91.8 | 32.8 | 49.2 | 68.5 | 76.2 | 39.4 | 59.1 | 77.0 | 87.0 | 20.2 | 59.0 | 73.1 | 78.2 |
| + Promter (Ours) | 0.41 | 20.20 | **48.3** | **74.2** | **88.0** | **90.7** | **27.3** | **34.6** | **60.9** | **70.2** | 30.3 | 66.8 | **87.3** | **92.2** | **34.1** | **49.5** | 67.8 | 74.3 | **43.8** | **60.4** | **77.9** | 86.5 | **21.3** | **59.5** | 73.0 | **78.3** |

# 4 EXPERIMENTS

## 4.1 EXPERIMENTAL SETUPS

**SSL Co-Training.** To apply the SSL task (*e.g.*, Prompter) to different base models for skeleton-based person re-ID, we employ the corresponding SSL objective (*e.g.*, $\mathcal{L}_{SSCR}$) as $\mathcal{L}_{SSL}$ in Eq. (2) to co-train the model. For person re-ID task, we leverage the learned model to encode raw skeleton sequences of the probe set $\Phi_P$ into feature representations ($\{V_i^P\}_{i=1}^{N_2}$) which are matched with the representations ($\{V_i^G\}_{i=1}^{N_3}$) in the gallery set $\Phi_G$ using Euclidean distance to predict the identity.

**Datasets.** We evaluate our method on four skeleton-based person re-ID datasets: *IAS* (Munaro et al., 2014b), *KS20* (Nambiar et al., 2017), *BIWI* (Munaro et al., 2014a), and *KGBD* (Andersson & Araujo, 2015), containing 11, 20, 50, and 164 different persons respectively. We also verify the generality of Prompter on RGB-estimated skeletons from a large-scale multi-view benchmark dataset *CASIA-B* (Yu et al., 2006) with 124 persons and three conditions (Normal (N), Bags (B), Clothes (C)). We adopt common probe and gallery settings for a fair comparison (Rao & Miao, 2023).

**Implementation Details.** We compare Prompter with different state-of-the-art SSL tasks (DR, MIC (Rao & Miao, 2022), STPR (Rao & Miao, 2023)) that can be compatibly co-trained with different state-of-the-art models. The number of different body joints is $J = 20$ in IAS, BIWI, KGBD, $J = 25$ in KS20, and $J = 14$ in the estimated skeletons of CASIA-B. The sequence length is $f = 6$ for

Table 4: Ablation study with different configurations: Probabilistic spatial (**PSCM**) or temporal context masking (**PTCM**). We also include random spatial masking (SM) or temporal masking (TM) with fixed mask numbers for performance comparison. "+" indicates employing the corresponding component, and "**+ PSCM + PTCM**" denotes the final configuration of Prompter.

| ID | Config. | KS20 | | IAS-A | | IAS-B | |
|---|---|---|---|---|---|---|---|
| | | mAP | $R_1$ | mAP | $R_1$ | mAP | $R_1$ |
| 1 | Baseline | 42.5 | 71.3 | 31.8 | 48.0 | 37.9 | 56.1 |
| 2 | + SM | 44.8 | 71.9 | 32.4 | 48.7 | 38.1 | 57.2 |
| 3 | + PSCM | 46.5 | 73.1 | 33.5 | 49.4 | 42.1 | 58.7 |
| 4 | + TM | 45.4 | 73.0 | 32.1 | 48.4 | 39.2 | 58.2 |
| 5 | + PTCM | 46.4 | 73.6 | 33.8 | 49.0 | 42.0 | 58.9 |
| 6 | + SM + TM | 46.2 | 73.6 | 32.8 | 49.2 | 39.4 | 59.1 |
| 7 | + PSCM + PTCM | 48.3 | 74.2 | 34.1 | 49.5 | 43.8 | 60.4 |

Table 5: Performance comparison of SSL tasks when applied to RGB-estimated skeletons on CASIA-B. ♣ refers to appearance-based methods. "B-N" represents using the "Bags (B)" probe set and "Normal (N)" gallery set. "—" indicates no published result.

| Probe-Gallery | C-C | | C-N | | B-N | | N-N | | B-B | |
|---|---|---|---|---|---|---|---|---|---|---|
| Methods | mAP | $R_1$ | mAP | $R_1$ | mAP | $R_1$ | mAP | $R_1$ | mAP | $R_1$ |
| LMNN♣ (Weinberger & Saul, 2009) | — | 17.4 | — | 11.6 | — | 23.1 | — | 3.9 | — | 18.3 |
| ITML♣ (Davis et al., 2007) | — | 20.1 | — | 10.3 | — | 21.8 | — | 7.5 | — | 19.5 |
| ELF♣ (Gray & Tao, 2008) | — | 19.9 | — | 5.6 | — | 17.1 | — | 12.3 | — | 5.8 |
| SDALF♣ (Farenzena et al., 2010) | — | 16.7 | — | 11.6 | — | 22.9 | — | 4.9 | — | 10.2 |
| MLR♣ (Scores) (Liu et al., 2015) | — | 13.5 | — | 9.7 | — | 14.7 | — | 13.6 | — | 13.6 |
| MLR♣ (Features) (Liu et al., 2015) | — | 25.4 | — | 20.3 | — | 31.8 | — | 16.3 | — | 18.9 |
| AGE (Rao et al., 2020) | 9.6 | 35.5 | 3.0 | 14.6 | 3.9 | 32.4 | 3.5 | 20.8 | 9.8 | 37.1 |
| SM-SGE (Rao et al., 2021a) | 9.7 | 27.2 | 3.0 | 10.6 | 3.5 | 16.6 | 6.6 | 50.2 | 9.3 | 26.6 |
| MG-SCR (Rao et al., 2021c) | 12.0 | 45.6 | 3.0 | 10.1 | 5.0 | 33.3 | 9.1 | 71.3 | 14.9 | 46.4 |
| SPC-MGR (Rao & Miao, 2022) | 11.8 | 48.3 | 4.3 | 22.4 | 4.6 | 28.9 | 9.1 | 71.2 | 11.4 | 44.3 |
| SGELA (Rao et al., 2021b) | 7.1 | 51.2 | 4.7 | 15.9 | 6.7 | 36.4 | 9.8 | 71.8 | 16.5 | 48.1 |
| + STPR (Rao & Miao, 2023) | 15.7 | 65.6 | **6.7** | 23.0 | 8.6 | 44.1 | 13.1 | 78.5 | **17.9** | 67.1 |
| + MIC | 16.1 | 67.8 | 5.6 | 22.2 | 8.9 | 43.0 | 13.1 | 76.2 | 16.5 | 64.0 |
| + DR | 14.0 | 64.2 | 4.9 | 20.4 | 8.1 | 44.2 | 13.5 | **85.0** | 17.5 | 65.0 |
| + Prompter (Ours) | **16.3** | **68.9** | **6.7** | **24.1** | **9.0** | **44.4** | **13.6** | 84.0 | 17.6 | 64.4 |

Kinect-based datasets (IAS, BIWI, KS20, KGBD) and $f = 40$ for the RGB-estimated skeleton data (CASIA-B), following existing methods for a fair comparison. We employ the MLP network with one hidden layer to build reconstructing and inferring models, and the embedding size is set to the same size of features used in the original base models. The probabilities for spatial and temporal masking are empirically set to $p_S = p_T = 0.5$, and we use $\alpha = 0.5$ to equally combine spatial and temporal reconstruction. We empirically adopt $\lambda = 0.5$ to fuse SSL and downstream task objectives. We report average performance under random parameter initializations following existing works for a fair comparison. More details are provided in the appendices.

**Evaluation Metrics.** The Cumulative Matching Characteristics (CMC) curve is calculated and we report Rank-1 ($R_1$), Rank-5 ($R_5$), and Rank-10 accuracy ($R_{10}$) as performance metrics. We also adopt Mean Average Precision (mAP) (Zheng et al., 2015) to evaluate the overall performance.

## 4.2 EMPIRICAL EVALUATION

**Performance of Prompter on State-of-the-Art Models.** As shown in Table 2, incorporating the proposed Prompter into SPC-MGR outperforms the original model without SSL by $0.7$-$18.1\%$ for Rank-1 accuracy and $0.8$-$4.0\%$ for mAP in most cases (10 of 12 cases) of different datasets. When applied to other state-of-the-art models without SSL, our approach also significantly improves their performance on *all* datasets by up to $7.3\%$ Rank-1 accuracy and $5.9\%$ mAP. This demonstrates the general effectiveness of Prompter on varying scenarios such as with frequent changes of viewpoints (KS20), occasions (BIWI-W), and appearances (IAS-A), and also verifies its strong compatibility with both supervised and unsupervised graph-based (MG-SCR, SPC-MGR, TranSG) and sequence-based models (SimMC) to learn more discriminative spatial-temporal skeleton semantics for person re-ID.

**Comparison with Different SSL Tasks.** Compared with existing SSL tasks, applying Prompter to different models achieves higher Rank-1 accuracy (23 of 24 cases) and mAP improvement (19 of 24 cases) in most datasets. It is interesting to note that MIC and DR usually produce large performance variations among models, which may suggest their inconsistent compatibility (CTC) under different models. Notably, our task also outperforms its transformable sub-tasks DR and STPR in most cases, which justifies our analysis that the SSL task combining different tasks can be more effective than the single contained task. With more consistent compatibility and higher effectiveness, Prompter can serve as a general SSL paradigm for skeleton-based person re-ID and more skeleton-related tasks.

**Ablation Study.** We evaluate the effectiveness of each component in Prompter and adopt TranSG without SSL (Rao & Miao, 2023) as the base model. As reported in Table 4, employing PSCM or PTCM (ID = 3, 5) achieves better performance than using direct spatial or temporal masking with fixed numbers of masks (ID = 2, 4) on different datasets. This verifies the effectiveness of the proposed probabilistic context masking, as it can generate more diverse random context of body structure and motion trajectory to facilitate reconstruction and the capture of richer useful semantics for person re-ID. Incorporating both PSCM and PTCM (ID = 7) further enhances the performance gain compared to direct masked reconstruction (ID = 6) (up to $4.4\%$ mAP and $1.3\%$ Rank-1 accuracy),

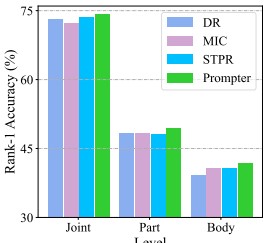
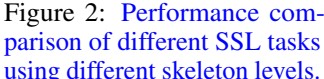

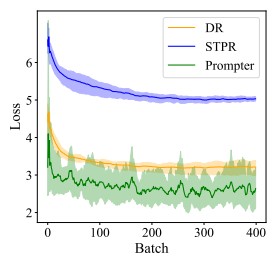

| Methods | BIWI→A | | BIWI→B | | IAS→W | | IAS→S | |
|---|---|---|---|---|---|---|---|---|
| | mAP | R₁ | mAP | R₁ | mAP | R₁ | mAP | R₁ |
| + STPR | 17.7 | 34.3 | 16.6 | 31.9 | 14.1 | 11.7 | 11.3 | 18.2 |
| + MIC | 18.2 | 34.4 | 16.4 | 34.2 | 14.0 | 11.6 | 11.4 | 19.9 |
| + DR | 17.7 | 31.3 | 18.2 | 32.8 | 15.1 | 12.0 | 11.6 | 18.8 |
| + Prompter (Ours) | 19.1 | 35.8 | 18.5 | 34.9 | 15.8 | 12.9 | 11.7 | 19.9 |

Figure 2: Performance comparison of different SSL tasks using different skeleton levels.

Figure 3: Losses of DR, STPR, and Prompter ($\mathcal{L}_{SSCR}$) when solely training Prompter.

Table 6: Performance comparison of different SSL tasks on the cross-domain person re-ID task. "A", "B", "W", "S" represent IAS-A, IAS-B, BIWI-W, BIWI-S. "IAS→W" denotes training the base model on the IAS training set and testing on BIWI-W.

which suggests the effectiveness and necessity of combining spatial and temporal skeleton semantics learning (*i.e.*, STE property) to learn more valuable distinguishing patterns for person re-ID.

### 4.3 FURTHER ANALYSIS

**Application to RGB-Estimated Skeletons.** To verify the general effectiveness of our method on estimated skeletons instead of Kinect-based skeletons, we extract 3D skeletons from CASIA-B using pose estimation models (Cao et al., 2019; Chen & Ramanan, 2017). As shown in Table 5, applying Prompter outperforms the latest SSL task STPR (Rao & Miao, 2023) by up to $5.5\%$ for Rank-1 accuracy and $0.6\%$ for mAP in four conditions, and it also achieves better performance than existing state-of-the-art skeleton-based models and classic appearance-based methods in most cases. This demonstrates the effectiveness of Prompter to facilitate learning richer valuable semantics from estimated skeletons, and further suggests its applicability to more general RGB-estimated scenarios.

**Evaluation on Cross-Domain Person Re-ID.** To validate the generality of skeleton semantics learned from Prompter, we co-train the base model with Prompter on the source datasets and evaluate its generalized performance on the target datasets without model fine-tuning. As shown in Table 6, applying Prompter achieves higher performance than using other SSL tasks when generalizing the learned model to other domains (*i.e.*, datasets), which suggests that our method could capture more general skeleton semantics (*e.g.*, domain-shared discriminative features) for person re-ID.

**Transfer to Different Skeleton Modeling.** We evaluate the effectiveness of transferring Prompter to varying levels of skeleton modeling (*e.g.*, part-level or body-level skeleton graphs (Rao et al., 2021a)). As shown in Fig. 2, our method outperforms different state-of-the-art SSL tasks on both original and higher-level skeleton representations. This demonstrates its generality and stronger effectiveness under different-level skeletal structures to facilitate the model to learn more discriminative features.

**Loss Visualization.** As shown in Fig. 3, solely applying Prompter simultaneously reduces DR and STPR losses, which validates its TT property that enables the model to jointly learn with contained sub-tasks. Consistent with the analysis in Sec. 3.3, Prompter introduces more diverse random cases into training (which could increase the fluctuations in loss) and can potentially reduce over-fitting to achieve a lower convergence value. More results and analyses are provided in the appendices.

## 5 CONCLUSION

In this paper, we propose the SCUT framework that identifies four key properties (STE, CTC, UT, TT) to assess the generality of skeleton semantics learning (SSL) tasks across different models and scenarios. Based on SCUT, we further devise a generic SSL task termed Prompter to probabilistically and independently mask spatial context of structural locations and temporal context of motion trajectories, which are exploited to reconstruct and infer complete skeleton sequences to capture general effective spatial-temporal skeleton semantics for person re-ID. Extensive evaluations on SCUT and five public datasets demonstrate the higher effectiveness and generality of Prompter than other state-of-the-art SSL tasks, and it is highly scalable to be applied to various models and scenarios.

ETHICS STATEMENT

We have reviewed the ICLR Code of Ethics and confirmed that our work complies with it.

Person re-ID models can be widely applied to different safe-critical areas such as security authentication, criminal tracking, and smart surveillance. However, illegally or irresponsibly deploying person re-ID technologies might invade personal privacy, thus it is important to establish relevant laws to protect the privacy. While existing skeleton-based person re-ID models do not disclose appearance-based information and have not been advanced enough to track individuals, such privacy issue should be kept in mind when developing this technology further (*e.g.*, combine with RGB images). Our models and codes must only be used for legitimate research.

We would like to emphasize that the datasets used in our work are officially shared by reliable research agencies, which guarantee that the collecting, processing, releasing, and using of data have gained the formal consent of participants. To protect privacy, each individual is anonymized with a simple identity number. We follow the official licenses of public datasets to assess and use skeleton data.

REPRODUCIBILITY STATEMENT

- Our anonymized codes and models are publicly available at https://github.com/Anonymous-9273/Prompter.

- In Sec. A of Appendix I, we provide details of experimental settings, including (1) Dataset description; (2) CASIA-B evaluation settings, (3) Dataset preprocessing strategy, (4) Probe/gallery settings, (5) Experimental setup details, and (6) Utilized computational resources.

- In Sec. B of Appendix I, we provide full experimental results for (1) Ablation study, (2) Effects of hyper-parameters, (3) Multi-shot performance with different sequence lengths $f$, and (4) Pseudo codes of Prompter.

- In Sec. C of Appendix I, we provide additional visualization and analysis of (1) Training metrics (*e.g.*, different losses), (2) Skeleton representations, and (3) Confusion metrics.

- In Sec. E of Appendix I, we provide additional experimental results and analyses based on reviewers' constructive comments and valuable suggestions, including:

  - We provide an additional comparison of key differences and similarity between our method (i.e., skeleton-based person re-ID) and skeleton-based gait recognition methods (for Reviewer iRXh).
  - We evaluate the performance of different state-of-the-art gait recognition methods (SkeletonGait, GaitTR, GPGait) on all datasets and compare them with our method (for Reviewer iRXh).
  - We provide an additional performance comparison of different SSL tasks (DR, MIC, STPR, Prompter) under different skeleton levels (Joint-Level, Part-Level, Body-Level) on different datasets (for Reviewer DvW6).
  - We offer qualitative examples and analyses for the cross-domain person re-ID performance, including confusion matrices and t-SNE feature visualization (for Reviewer DvW6).
  - We provide an additional overview of state-of-the-art skeleton semantics learning (SSL) tasks, their source method, and method types (for Reviewer v2zj).
  - We offer a detailed comparison between our method and existing state-of-the-art masking strategies (for Reviewers iRXh, BHkC, v2zj).
  - We integrate the proposed Prompter into the representative state-of-the-art gait recognition method GPGait, and compare its performance with the original base model on different datasets (for Reviwer DUn6).
  - We additionally evaluated the representative state-of-the-art gait recognition method and action recognition method ST-GCN on our benchmark datasets, and integrated the proposed Prompter into them to verify its general applicability (for Reviewers BHkC, iRXh, DvW6).

- In Sec. A of Appendix II, we offer a general computing formula for occurrence probabilities of different sub-tasks contained in Prompter.
- In Sec. B of Appendix II, we provide theoretical assumptions and analyses of Prompter on potential model regularization.

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
