# OpenReview forum: "General Skeleton Semantics Learning with Probabilistic Masked Context Reconstruction for Skeleton-Based Person Re-Identification"
_ICLR.cc/2025/Conference — Submitted to ICLR 2025_

### Official Review · Reviewer_v2zj · 2024-10-28

**Soundness:** 3
**Presentation:** 3
**Contribution:** 4
**Rating:** 8
**Confidence:** 4

**Summary:**

This paper proposes a new generality assessment framework and a probabilistic masked spatial-temporal reconstruction method. Specifically, the assessment framework encompasses four properties (CTC, STE, UT, TT) from quantitative and qualitative perspectives to evaluate the generality of methods. Then, the probabilistic masked strategies are utilized in the skeleton at the spatial and temporal levels and reconstructed to learn task-agnostic skeleton representation. Extensive experiments show the method achieves superior results.

**Strengths:**

1. The idea of this paper is exciting, especially in the assessment framework aspect.
2. The proposed assessment framework should be appreciated, as it can be helpful in the skeleton representation learning field.
3. The experimental results achieve the sota.

**Weaknesses:**

1. The indicator used in the assessment framework lacks a reasonable explanation.
2. The proposed assessment framework is not evaluated in the supervised/self-supervised skeleton representation learning methods.

**Questions:**

1. I have a doubt concerning the quantitative properties. The CTC and STE utilize the Rank-1 accuracy as the performance indicator. However, the skeleton semantic representation should be evaluated at the feature level, not the logits of the output. Why did the authors choose to use this indicator? I think some feature cluster metrics, such as feature silhouette sore and FMI score, may be more reasonable.
2. The authors mentioned that their method has unsupervised trainability. Therefore,  the authors should evaluate the zero-shot capability of their method as they have described in line 235 that the UT can guarantee the SSL task applies to unknown classes.
3. The mask-reconstruction pipeline has been employed in many studies; what motivates using probabilistic masking in this method rather than fixed ratio mask strategies such as MAE? Meanwhile, what is necessary to leverage them into spatial and temporal levels, respectively, should be described clearly.
4. Does this assessment framework only suit the task of person re-identification? The authors should add the experiments to evaluate them in the supervised/self-supervised skeleton representation learning methods.

---

> ### Author Response · Authors · 2024-11-26
>
> We thank you for recognizing the novelty and contributions of our paper and also for the positive feedback. We would like to answer your questions below.
>
> >**Q1**. ”The indicator used in the assessment framework lacks a reasonable explanation.”
>
> **A1**. Thank you for your comment. We have provided detailed motivation and evaluation criteria for all properties defined in the generality assessment framework, including Co-training compatibility (CTC) (motivation explained in Line 171-176, Line 184-193), Spatial-temporal effectiveness (STE) (motivation explained in Line 194-200, Line 210-233), Unsupervised trainability (UT) (motivation explained in Line 236-239), Task transformability (TT) (motivation explained in Line 244-249, further analysis in Line 367-377 and Sec. 4.3).
>
> >**Q2**. ”The proposed assessment framework is not evaluated in the supervised/self-supervised skeleton representation learning methods.”
>
> &
>
> >**Q3**. ”The authors should add the experiments to evaluate them in the supervised/self-supervised skeleton representation learning methods.”
>
> **A2** & **A3**. Thank you for your valuable suggestion. We would like to clarify that seven existing skeleton semantics learning (SSL) tasks used in the supervised (e.g., TranSG and its task STPR), self-supervised (e.g., SM-SGE and its task MSR, AGE and its task AR), and unsupervised skeleton-based person re-ID methods (e.g., SimMC and its task MIC) have been evaluated (please see Table 1, Table 2, and Table 3, Line 378-381). To present a more clear category of different SSL tasks and their corresponding methods, we provide an overview table (corresponding to Table 1 in our paper) for reviewers as follows (highlighted in our Appendix I Sec. E.3)):
>
> | ID |     SSL Task    | Source Method |          Method Type         |
> |:--:|:---------------:|:-------------:|:----------------------------:|
> |  1 |       DR        |       —       |               —              |
> |  2 |        AR       |      AGE      |        Self-Supervised       |
> |  3 |     AR + AC     |     SGELA     |        Self-Supervised       |
> |  4 |      MSSP       |     MG-SCR    |          Supervised          |
> |  5 |       MSR       |     SM-SGE    | Supervised / Self-Supervised |
> |  6 |       MIC       |     SimMC     |         Unsupervised         |
> |  7 |      STPR       |     TranSG    |          Supervised          |
> |  8 | Prompter (Ours) |       —       |               —              |

---

> ### Author Response · Authors · 2024-11-26
>
> >**Q4**. ”I have a doubt concerning the quantitative properties. The CTC and STE utilize the Rank-1 accuracy as the performance indicator. However, the skeleton semantic representation should be evaluated at the feature level, not the logits of the output. Why did the authors choose to use this indicator?”
>
> &
>
> >**Q5**. ”I have a doubt concerning the quantitative properties. The CTC and STE utilize the Rank-1 accuracy as the performance indicator. However, the skeleton semantic representation should be evaluated at the feature level, not the logits of the output. Why did the authors choose to use this indicator? I think some feature cluster metrics, such as feature silhouette sore and FMI score, may be more reasonable.”
>
> **A4** & **A5**. Thanks for your valuable question.
> - We would like to point out that all properties defined in the generality assessment framework aim to evaluate the overall generality (e.g., training compatibility with different models under varying datasets) of skeleton semantics learning (SSL) tasks, instead of its single-dimensional effectiveness (e.g., feature-level accuracy related to logits/output). In fact, the co-training compatibility (CTC) or spatial-temporal effectiveness (STE) may serve as a qualitative attribute for evaluation, e.g., based on whether the SSL task can be co-trained with different models and whether it possesses both spatial and temporal semantics learning. However, considering that such evaluation may be vague and subjective, we further propose concrete computation formula based on performance gain specifically for these properties to quantitatively measure them (please see Sec. 3.2). It is worth noting that the performance gain is measured by the most frequently used metric, Rank-1 accuracy (please see Line 187-189), which can provide an intuitive indicator of performance improvement when employing an SSL task on different models and datasets. In particular, we choose Rank-1 accuracy metric due to its intuitiveness (i.e., intuitively measure model performance improvement), simplicity (i.e., easier computation than other metrics like mAP), and uniformity (i.e., all existing skeleton-based person re-ID employ this unified metric) (we will add and highlight this in our paper). To better understand the role of performance gain (Rank-1 accuracy), CTC, and STE proposed in our framework, we provide the following explanation:
>
> - In G_C (CTC), we focus on the average co-training performance gain under different models and datasets, so we incorporate the respective gain of a certain model on a specific dataset with corresponding weight coefficient (please see Eq. (3), Line 181-189, Sec. 3.2), so as to measure whether and how much performance gain in average that an SSL task can compatibly facilitate on different combinations of models and datasets. Different from G_C (CTC) that computes the overall performance gain, the proposed spatial-temporal effectiveness (STE), i.e., G_ST, aims to quantify the average spatial-temporal performance gain. It contains a unique component R^{ST}_{i, j} to compute the relative ratio between the performance gains of spatial part and temporal part of an SSL task (please see Eq. (5), Line 209-230, Sec. 3.2). This component not only considers independent performance gain from spatial or temporal component (e.g., spatial/temporal semantics learning) of an SSL task, but also synergizes their contribution balance to encourage a more reasonable measurement of spatial-temporal effectiveness (please see definition and motivation in Line 213-233).
>
> - To simultaneously evaluate the overall co-training performance gain as an indicator of SSL compatibility (CTC), and measure the combined effectiveness of independent spatial and temporal parts of SSL to indicate the spatial-temporal modeling effectiveness (STE) of SSL, both G_ST and G_C are necessary in the proposed SSL generality assessment framework.
>
> Thanks again for the inspiring comments of the reviewer, and we will further explore other metrics in our future direction.

---

> ### Author Response · Authors · 2024-11-26
>
> >**Q6**. ”The authors mentioned that their method has unsupervised trainability. Therefore, the authors should evaluate the zero-shot capability of their method as they have described in line 235 that the UT can guarantee the SSL task applies to unknown classes.”
>
> **A6**. Our method can be trained in an unsupervised manner, and can be widely applied to different kinds of methods. In our work, we have evaluated the proposed methods on not only unsupervised methods (e.g., SPC-MGR, SimMC) but also self-supervised/supervised methods (e.g., MG-SCR, TranSG) (please see Sec. 4.2). Note that our method Prompter does not require any labels to train on the above different methods to improve their performance in most cases, and could also achieve better performance than most existing hand-crafted, supervised, self-supervised, unsupervised methods on different datatsets (please see Table 2, 3, and 5, Sec. 4.2). Additionally, we also investigate and verify the cross-domain person re-ID effectiveness of our method (i.e., zero-shot/generalized capability on a different dataset) on different datasets (please see Table 6, Sec. 4.3).
>
>
> >**Q7**. ”The mask-reconstruction pipeline has been employed in many studies; what motivates using probabilistic masking in this method rather than fixed ratio mask strategies such as MAE? Meanwhile, what is necessary to leverage them into spatial and temporal levels, respectively, should be described clearly.”
>
>
> **A7**. Thanks for your comment. We have added detailed comparsion in Appendix I Sec. E. 3.1.
> - Firstly, compared with existing spatial-temporal masking strategies such as SkeletonMAE [1] and MS2L [2], we hope to clarify that the key novelty of the proposed Probabilistic Spatial Context Masking (PSCM) and Probabilistic Temporal Context Masking (PTCM) is that they are devised at an independent level of body structural locations and motion trajectory positions based on independent and identically distributed (IID) Bernoulli random masks. It possesses higher generality than previous methods (detailed in Sec. 3.2) and can be probabilistically generalized to different existing masking mechanisms for more effective skeleton semantics learning (please see TT property and Line 367-377). By contrast, existing masking strategies such as MS2L [2] directly mask the later consecutive skeletons (i.e., 150 frames) for temporal predictions while failing to learn effective spatial relations (i.e., performance degrades) under the used spatial masking. In [1], the structural positions are masked conditioned on the temporally-masked frames. Such direct frame-level or conditioned masking has several limitations, such as they cannot explicitly and individually model effective spatial semantics, nor can it feasibly evaluate the performance contribution of spatial masking (please refer to STE property defined in our work), while our method has solved these challenges with a focus of more generalizable skeleton context reconstruction.
> - Secondly, in our experiments, we also systematically compare our method with state-of-the-art SSL tasks using different masking strategies: Direct temporal masking (MIC), random masking with fixed-number masks (STPR), and the baseline without masking (DR). The experimental results demonstrate the higher effectiveness of our approach that adopts independent and finer-grained spatial-temporal masking. Moreover, we hope to highlight another novel contribution of our work is to propose a systematic SSL generality assessment framework (SCUT) to explore the multi-faceted performance and bottlenecks of existing SSL tasks under varying models and scenarios (please see Line 58-76). Motivated by the identified key properties of SCUT, we focus on devising a general solution (Prompter) that can be flexibly applied to different state-of-the-art skeleton-based person re-ID models (e.g., graph transformer, GAT, Siamese encoders). This is fundamentally different from MS2L [1] and [2] that rely on a certain action recognition backbone or model (i.e., GRU or STTFormer) to design effective masking strategies. Therefore, our method could be more general and scalable than these methods. Prompter can also be flexibly applied to RGB-estimated skeletons, unsupervised scenarios, different graph modeling, and cross-domain person re-ID tasks (see Further Analyses in our paper).
>
> References:
>
> [1] Wu, Wenhan, et al. "Skeletonmae: Spatial-temporal masked autoencoders for self-supervised skeleton action recognition." 2023 IEEE international conference on multimedia and expo workshops (ICMEW), IEEE, 2023.
>
> [2] Lin, Lilang, et al. "Ms2l: Multi-task self-supervised learning for skeleton based action recognition." Proceedings of the 28th ACM international conference on multimedia, 2020.
>
> We thank you for appreciating our contributions. We sincerely hope our clarifications above have addressed your concerns and can improve your opinion of our work.

---

> > ### Author Response · Authors · 2024-11-28
> > **Thanks to Reviewer v2zj**
> >
> > Dear Reviewer v2zj,
> >
> > We would like to thank you again for your valuable feedback to improve our work. We are wondering whether our response has addressed your questions and can improve your opinion of our work.
> >
> > Kindly let us know if your might have further comments, and we will do our best to address them.
> >
> > Best Regards,
> >
> > The Authors

---

> > ### Comment · Reviewer_v2zj · 2024-11-29
> >
> > Thanks for your detailed response.
> >
> > All my concerns have been addressed successfully, except for the aspect of zero-shot capability. I noticed that the authors evaluate the cross-domain transferability across multiple datasets in Table 6, which is comprehensive and convincing and should be appreciated. However, this setting does not entirely align with the conventional zero-shot evaluation criteria (i.e., recognizing novel/unseen classes). Although cross-domain evaluation and zero-shot setting have similar aspects, an important distinction remains: cross-domain focuses on recognizing unseen domains, whereas zero-shot is concerned with unseen categories. For example, in the skeleton-based action recognition field, the cross-view and cross-subject settings are typically utilized to assess the generalizability of methods. Nevertheless, neither of them meets the zero-shot criteria. More specifically, in the zero-shot setting, the method should be trained on seen categories (e.g., drinking, reading) and tested on unseen categories (e.g., writing, eating). Thus, I encourage the author to clarify this point more specifically and precisely, not misleading the readers.
> >
> > Furthermore, I would like to express my sincere appreciation for the contributions of the proposed generality assessment metrics. I believe that this unified framework has the potential to advance the development of this field, not only in the skeleton-based person re-identification area but also in the whole skeleton-based research community.
> >
> > Overall, I am willing to raise my score based on the responses of the authors and their contributions to this work.

---

> ### Author Response · Authors · 2024-11-29
> **Thank You for Appreciating Our Work & Increasing Your Score & Further Response**
>
> Dear Reviewer,
>
> We are glad to know that our response has successfully addressed your previous questions. We would like to thank you again for appreciating our work and upgrading your score! We would like to answer your further questions below.
>
> >**Q1**. ”However, this setting does not entirely align with the conventional zero-shot evaluation criteria (i.e., recognizing novel/unseen classes). Although cross-domain evaluation and zero-shot setting have similar aspects, an important distinction remains: cross-domain focuses on recognizing unseen domains, whereas zero-shot is concerned with unseen categories. “
>
> &
>
> >**Q2**. ”More specifically, in the zero-shot setting, the method should be trained on seen categories (e.g., drinking, reading) and tested on unseen categories (e.g., writing, eating). Thus, I encourage the author to clarify this point more specifically and precisely.”
>
> **A1** & **A2**. Thank you for your valuable comment. We hope to clarify that the “cross-domain” evaluation in our work essentially follows the “zero-shot” evaluation criteria (i.e., recognizing novel/unseen classes). In particular, we train the model on the source datasets (e.g., BIWI training set) and evaluate its generalized performance on the target datasets (e.g., IAS-A) without model fine-tuning (please see Line 513-518 of our paper and Line 359-362 of Appendix I). It is worth noting that **the persons/pedestrians in the source dataset and target dataset are totally different** (i.e., the classes in IAS-A are novel/unseen classes different from BIWI classes) although they may have the same ID number (i.e., the person with ID=1 in BIWI (BIWI RGBD-ID) is different from the person with ID=1 in IAS-A (IAS-Lab RGBD-ID), please refer to the description of two different datasets at https://robotics.dei.unipd.it/reid/index.php). Our method can be trained on the source dataset and tested on the target dataset, without requiring any training or testing data from the target dataset to re-train the model (i.e., without model fine-tuning or adaptation). **As the testing set (i.e., target dataset) only contains the unseen classes that are different from the source dataset, this evaluation aligns with the zero-shot protocol**. The results in Table 6 demonstrate the higher effectiveness of the proposed Prompter than other SSL tasks when generalizing the learned model to other unseen datasets, suggesting that our method could capture more general skeleton semantics for person re-ID. Moreover, we have provided additional qualitative analyses for the generalized performance across datasets, including confusing matrices for analyzing accuracy differences on different classes, and t-SNE feature visualization for analyzing the intra-class and inter-class similarity among classes (please see Appendix I Sec. E.2, Fig. 11, 12, 13).
>
> We sincerely apologize for the confusion caused by the term “cross-domain” (which we intended to highlight generalized person re-ID across datasets), and will revise it to a more specific term (e.g., “zero-shot cross-dataset evaluation”) to more precisely describe the current evaluation setting in the future version of our paper.
>
> We sincerely hope our clarifications above have addressed your questions. We would like to thank you again for appreciating our work and recognizing our contributions!
>
> Best Regards,
>
> The Authors

---

> > ### Comment · Reviewer_v2zj · 2024-11-29
> >
> > Thanks for your clear clarifications. All my concerns have been addressed.

---

> > > ### Author Response · Authors · 2024-11-29
> > > **Thank You for Appreciating Our Response!**
> > >
> > > Dear Reviewer,
> > >
> > > We are glad to know that our response has addressed all of your questions. We would like to thank you again for appreciating our work and recognizing our contributions!
> > >
> > > Best Regards,
> > >
> > > The Authors

---

### Official Review · Reviewer_DUn6 · 2024-10-29

**Soundness:** 3
**Presentation:** 3
**Contribution:** 3
**Rating:** 8
**Confidence:** 5

**Summary:**

The paper addresses the task of Skeleton Semantics Learning (SSL) with two contributions. First, it proposes a framework to assess the generality of a SSL model with 4 criteria: Spatial-temporal effectiveness, Co- training compatibility, Unsupervised trainability, and Task transformability. Second, it proposes Prompter, a novel idea that can be integrated into any SSL model to enhance the discriminative power of extracted skeleton features, where spatial and temporal context are masked and then reconstructed and inferred.

**Strengths:**

- The proposed framework for assessing the generality of a SSL model is a great contribution. The 4 suggested criteria are reasonable and can be applied to many other tasks as well.
- The proposed idea of probabilistically and independently mask the spatial and temporal context of skeletons and then reconstruct and infer the sequence, though sounds simple, is reasonable and shows great improvement when being coupled with previous SSL methods.
- Valuable aspects of the framework are discussed and analyzed, including Loss Visualization, Cross-domain evaluation, ...
- Writing is clear and technical details are correctly presented.

**Weaknesses:**

- References are not adequate. Literature in Skeleton-Based Person Re-Identification is quite richer than what has been covered in the paper. For example, see the papers below.
- The chosen datasets are specifically designed for SSL, while the paper claims the paper contributes significantly to Person Re-ID, thus, experiments on Re-ID datasets should be conducted to support the claim and make the paper more convincing.
- Results comparison are not comprehensive enough, where barely any direct Re-ID method is taken into consideration.
- SSL should be widely applied in Gait Recognition task but not limited to Person Re-Identification. Thus, results comparison or results of integration of Prompter into Gait Recognition method would be necessary.

[1] Wang et al., Human skeleton mutual learning for person re-identification. In Neurcomputing, 2021.
[2] Lu et al., Exploring high-order spatio–temporal correlations from skeleton for person Re-identification. In TIP, 2023.
[3] Nguyen et al., Attention-based shape and gait representations learning for video-based cloth-changing person re-identification. In VISIGRAPP, 2024.

**Questions:**

Please see the weaknesses, which also include my main questions/concerns.

---

> ### Author Response · Authors · 2024-11-26
>
> We deeply appreciate your positive feedback and constructive comments on improving our paper. We will address your questions below.
>
> >**Q1**. ”References are not adequate. Literature in Skeleton-Based Person Re-Identification is quite richer than what has been covered in the paper. For example, see the papers below.”
>
>
> **A1**. Thank you for your valuable suggestions. We have added these references in our related works for a brief introduction (highlighted in blue). However, we would like to clarify that the works [1]-[3] may not be DIRECTLY related to our topic, as they mainly focus on RGB image/videos based person re-identification (re-ID) and use estimated skeletons from them as auxiliary features to enhance the performance. In contrast, the focus of skeleton-based person re-ID is to utilize ONLY 3D skeleton data to identify different persons, and this task typically uses 3D skeletons captured from depth sensors (e.g., the public benchmark datasets IAS, KS20, BIWI, KGBD used in our work (please see Line 425-430)). This is significantly different from [1]-[3] that adopt RGB images as inputs and use the skeleton data (generally 2D skeletons) estimated from RGB videos (e.g., RGB-based datasets MARS, PRID-2011 that originally do NOT contain any skeleton data), which typically possesses different skeletal topologies and contains much more noise influenced by the quality of pose estimation models. Moreover, compared with [1]-[3] that focus on the challenges of clothing variations, appearance changes, background clutters, or body partition/segmentation in RGB-based person re-ID, our work concentrates on devising a general skeleton semantics learning (SSL) task to be applied to different models to learn effective spatial-temporal skeleton semantics to improve person re-ID performance (please see Line 144-161, Sec. 3.1). Therefore, they could belong to different research problems. Considering that the inherent differences in input data modalities, skeleton data types, focused problems, and evaluation settings may cause an unfair comparison, we do not include these methods in our main part.
>
>
>
> >**Q2**. ”The chosen datasets are specifically designed for SSL, while the paper claims the paper contributes significantly to Person Re-ID, thus, experiments on Re-ID datasets should be conducted to support the claim and make the paper more convincing.”
>
> &
>
> >**Q3**. ”Results comparison are not comprehensive enough, where barely any direct Re-ID method is taken into consideration.”
>
> **A2** & **A3**. Thank you for your comments, and we will improve our experimental comparison in our future version. We would like to clarify that the chosen four public datasets (IAS, KS20, BIWI, KGBD), which contain Kinect-based 3D skeleton data, are commonly used for performance evaluation of skeleton-based person re-ID methods [4], i.e., skeleton-based person re-ID benchmark datasets, which are NOT specifically designed for SSL only. We also include a large-scale multi-view benchmark dataset CASIA-B to verify the generality of our method on RGB-estimated skeletons. We follow previous 11 state-of-the-art skeleton-based person re-ID methods (AGE, SGELA, MG-SCR, SimMC, TranSG, etc., please see Table 2) and 3 state-of-the-art SSL tasks (please se Table 3) to conduct experiments with the same evaluation setting on these four benchmark datasets that contain sensor-based 3D skeletons for a fair comparison. We further compare the generality and performance of our method with 6 classic appearance-based person re-ID methods, 5 state-of-the-art skeleton-based person re-ID methods, and 3 state-of-the-art SSL tasks (please see Table 5) on RGB-estimated skeletons (CASIA-B).
>
> References:
>
> [1]Wang et al., Human skeleton mutual learning for person re-identification. In Neurcomputing, 2021.
>
> [2] Lu et al., Exploring high-order spatio–temporal correlations from skeleton for person Re-identification. In TIP, 2023.
>
> [3] Nguyen et al., Attention-based shape and gait representations learning for video-based cloth-changing person re-identification. In VISIGRAPP, 2024.
>
> [4] Rao et al. TranSG: Transformer-Based Skeleton Graph Prototype Contrastive Learning with Structure-Trajectory Prompted Reconstruction for Person Re-Identification. In CVPR, 2023.

---

> > ### Comment · Reviewer_DUn6 · 2024-11-27
> >
> > Thank you for your clarification about SSL-based Person Re-ID. My concerns are almost addressed.

---

> ### Author Response · Authors · 2024-11-27
>
> >**Q4**. ”SSL should be widely applied in Gait Recognition task but not limited to Person Re-Identification. Thus, results comparison or results of integration of Prompter into Gait Recognition method would be necessary.”
>
> **A4**. Thanks for your constructive and insightful comment. We integrate our proposed SSL task into the representative state-of-the-art gait recognition method GPGait [1] and evaluate its performance compared with the original model. As shown in the table below (also added in Appendix E.4, Table 13), our proposed Prompter consistently improves the performance (mAP) of GPGait on different datasets, which demonstrates the effectiveness of Prompter (i.e., the proposed spatial-temporal skeleton semantics learning) when applied to the state-of-the-art gait recognition method. This further suggests the generality of our SSL task to be applied to different related areas.
>
>
> | Method            | KS20 | BIWI-W | BIWI-S | IAS-A | IAS-B | KGBD |
> |-------------------|:----:|:------:|:------:|:-----:|:-----:|:----:|
> | GPGait            | 41.1 |   25.5 |   23.5 |  34.6 |  43.1 | 17.8 |
> | + Prompter (Ours) | **43.3** |   **27.2** |   **24.7** |  **37.5** |  **45.6** | **18.6** |
>
> References:
>
> [1] Yang, et al. Gpgait: Generalized pose-based gait recognition. Proceedings of the IEEE/CVF International Conference on Computer Vision. 2023.
>
> We thank you for appreciating our contributions. We sincerely hope our clarifications above have addressed your concerns and can improve your opinion of our work.

---

> > ### Comment · Reviewer_DUn6 · 2024-11-27
> >
> > Thank you for providing more experimental results in coupling Prompter with a gait recognition model. The results look consistent and convincing.

---

> ### Author Response · Authors · 2024-11-27
> **Thank You for Appreciating Our Response!**
>
> Dear Reviewer,
>
> We are glad to know that our response has addressed your questions. We would like to thank you again for appreciating our work and recognizing our contributions!
>
>
> Best Regards,
>
> The Authors

---

### Official Review · Reviewer_iRXh · 2024-11-01

**Soundness:** 2
**Presentation:** 2
**Contribution:** 2
**Rating:** 6
**Confidence:** 3

**Summary:**

The paper addresses person re-identification (re-ID) using skeleton-based methods, which offer advantages like smaller data input and better privacy protection. Traditional methods rely on visual features, while skeleton-based approaches focus on body structure and motion semantics. The authors introduce a systematic generality assessment framework called SCUT to evaluate self-supervised learning (SSL) tasks. They propose a novel SSL task, Prompter, which uses probabilistic spatial and temporal context masking to enhance skeleton sequence reconstruction. The study highlights the importance of spatial-temporal effectiveness and model compatibility in SSL, demonstrating Prompter’s effectiveness across various datasets and models.

**Strengths:**

1. The authors identify four general assessment methods for evaluating skeleton semantic learning, which brings new insights into this topic.
2. The extensive experiments demonstrate that the proposed Prompter achieves better results over different model architectures and benchmarks.

**Weaknesses:**

1. Since skeleton-based person re-ID is fundamentally similar to skeleton-based gait recognition, the authors should compare their work with more recent gait studies in terms of method design and accuracy, such as SkeletonGait [1], GPGait [2], and CAG [3]. This comparison is crucial to assess the superiority of their approach.

2. The proposed Prompter technique, which involves spatial-temporal masking for skeleton sequences, has been extensively explored in previous works like SkeletonMAE [4] and MS2L [5]. These methods have studied masking skeleton joints in spatial and temporal dimensions. Consequently, the novelty of this masking technique appears limited.

[1] Fan, Chao, et al. "SkeletonGait: Gait Recognition Using Skeleton Maps." Proceedings of the AAAI Conference on Artificial Intelligence. Vol. 38. No. 2. 2024.

[2] Fu, Yang, et al. "Gpgait: Generalized pose-based gait recognition." Proceedings of the IEEE/CVF International Conference on Computer Vision. 2023.

[3] Huang, Xiaohu, et al. "Condition-adaptive graph convolution learning for skeleton-based gait recognition." IEEE Transactions on Image Processing (2023).

[4] Wu, Wenhan, et al. "Skeletonmae: Spatial-temporal masked autoencoders for self-supervised skeleton action recognition." 2023 IEEE international conference on multimedia and expo workshops (ICMEW). IEEE, 2023.

[5] Lin, Lilang, et al. "Ms2l: Multi-task self-supervised learning for skeleton based action recognition." Proceedings of the 28th ACM international conference on multimedia. 2020.

**Questions:**

1. The recent skeleton-based gait recognition papers should be cited and compared, as they are strongly related to this work (see weaknesses).

2. The proposed masking method has limited novelty since some relevant papers have studied this topic extensively (see weaknesses).

---

> ### Author Response · Authors · 2024-11-24
>
> >**Q1**. Since skeleton-based person re-ID is fundamentally similar to skeleton-based gait recognition, the authors should compare their work with more recent gait studies in terms of method design and accuracy, such as SkeletonGait [1], GPGait [2], and CAG [3]. This comparison is crucial to assess the superiority of their approach.
>
> &
>
> >**Q2**. The recent skeleton-based gait recognition papers should be cited and compared, as they are strongly related to this work (see weaknesses).
>
> **A1** & **A2**. Thanks for your valuable comment.
>
> - Firstly, although [1]-[3] and our work are related to gait topics, we would like to point out that skeleton-based person re-ID differs from skeleton-based gait recognition, in terms of their tasks, input data types (3D skeletons (25, 20, or 14 joints) VS. 2D skeletons (typically 17 joints)), application scenarios (Sensor/Model-estimated skeletons VS. Model-estimated skeletons), learning scenarios (Supervised/Unsupervised VS. Supervised), etc. To better compare the inherent similarity and differences between our work and these gait recognition works [1]-[3], we provide an additional comparison table as follows:
>
> | Method                |                     Prompter (Ours)                    |              [1]-[3]              |
> |-----------------------|:------------------------------------------------------:|:---------------------------------:|
> | Task                  |                Person Re-Identification                |          Gait Recognition         |
> | Focused Problem       |                 Matching and Retrieving; Classification                |           Matching and Retrieving; Classification          |
> | Input Skeleton Type   |      3D (Support 25, 20 or 14 joints in our work)      |      2D (Typically 17 joints)     |
> | Application Scenarios |   Sensor-based skeletons;   Model-estimated skeletons  |     Model-estimated skeletons     |
> | Base Architectures    | Support different models (e.g., Transformer, GAT, MLP) |         Rely on GCN or CNN        |
> | Datasets              |                Five 3D skeleton datasets               | Two to Five 2D skeleton datasets  |
> | Learning Scenarios    |           Support supervised and unsupervised          |          Only Supervised          |
> | Skeletal Topology     |                 Support multiple levels                |         Only single level         |
>
> - Secondly, we provide a performance and efficiency comparison between our approach (best setting with TranSG), and SkeletonGait [1] (skeletons used), GPGait [2], and GaitTR [6] with optimal parameter settings (Note that [3] is not compared here as its official source code is still unavailable). We adopt the same evaluation protocol (i.e., same datasets, skeletal topology, joint number, limb/bone partition, etc.) used in our work for a fair comparison. As shown in the table below, our method significantly outperforms them with less parameter size and lower computational complexity in most cases, which demonstrates the superiority of our approach. As these three methods are typically designed for 2D skeleton based gait recognition tasks, these results imply that a more appropriate and efficient GCN or CNN architecture could be further designed specifically for 3D skeleton based person re-ID. Considering that the inherent gaps (please refer to (1)) between these methods and our method might influence their performance on a different task to cause unfair comparison (e.g., poor performance of SkeletonGait on person re-ID), the provided additional experimental results under default optimal parameter settings are only for a performance reference.
>
> |  Method                 |  # Paras |  GFLOPs | KS20 |        | BIWI-W |        | BIWI-S |        | IAS-A |        | IAS-B |        | KGBD |        |
> |-------------------------|---------:|--------:|:----:|--------|:------:|--------|:------:|:------:|:-----:|--------|:-----:|--------|:----:|--------|
> |                         |          |         |  mAP | Rank-1 |    mAP | Rank-1 |    mAP | Rank-1 |   mAP | Rank-1 |   mAP | Rank-1 |  mAP | Rank-1 |
> | SkeletonGait (Skeleton) |   11.11M | 1592.60 | 14.5 |   22.2 |   11.2 |   10.8 |    9.3 |   15.1 |  25.0 |   31.4 |  21.5 |   31.5 |  1.1 |    1.7 |
> | GaitTR                  |    0.49M |   18.20 | 25.5 |   52.4 |   19.8 |   21.9 |   18.8 |   46.3 |  31.1 |   44.2 |  34.2 |   49.4 | 13.5 |   51.6 |
> | GPGait                  |    1.30M |   49.62 | 41.1 |   71.4 |   25.5 |   29.0 |   23.5 |   54.1 |  34.6 |   50.9 |  43.1 |   60.1 | 17.8 |   53.9 |
> | Prompter (Ours)         |    0.41M |   20.20 | 48.3 |   74.2 |   27.3 |   34.6 |   30.3 |   66.8 |  34.1 |   49.5 |  43.8 |   60.4 | 21.3 |   59.5 |
>
> We also added these two tables to the revised supplementary material (please see Appendix I Sec. E.1).

---

> ### Author Response · Authors · 2024-11-24
>
> >**Q3**. The proposed Prompter technique, which involves spatial-temporal masking for skeleton sequences, has been extensively explored in previous works like SkeletonMAE [4] and MS2L [5]. These methods have studied masking skeleton joints in spatial and temporal dimensions. Consequently, the novelty of this masking technique appears limited.
>
> &
>
> >**Q4**. The proposed masking method has limited novelty since some relevant papers have studied this topic extensively (see weaknesses).
>
> **A3** & **A4**. Thanks for your valuable feedback.
>
> - Firstly, compared with [4]-[5], we hope to clarify that the key novelty of the proposed Probabilistic Spatial Context Masking (PSCM) and Probabilistic Temporal Context Masking (PTCM) is that they are devised at an independent level of body structural locations and motion trajectory positions based on independent and identically distributed (IID) Bernoulli random masks. It possesses higher generality than previous methods (detailed in Sec. 3.2) and can be probabilistically generalized to different existing masking mechanisms for more effective skeleton semantics learning (please see TT property and Line 402-413). By contrast, MS2L [5] directly masks the later consecutive skeletons (i.e., 150 frames) for temporal predictions while failing to learn effective spatial relations (i.e., performance degrades) under the used spatial masking. In [6], the structural positions are masked conditioned on the temporally-masked frames. Such direct frame-level or conditioned masking has several limitations, such as they cannot explicitly and individually model effective spatial semantics, nor can it feasibly evaluate the performance contribution of spatial masking (please refer to STE property defined in our work), while our method has solved these challenges with a focus of more generalizable skeleton context reconstruction.
>
> - Secondly, in our experiments, we also systematically compare our method with state-of-the-art SSL tasks using different masking strategies: Direct temporal masking (MIC), random masking with fixed-number masks (STPR), and the baseline without masking (DR). The experimental results demonstrate the higher effectiveness of our approach that adopts independent and finer-grained spatial-temporal masking. Moreover, we hope to highlight another novel contribution of our work is to propose a systematic SSL generality assessment framework (SCUT) to explore the multi-faceted performance and bottlenecks of existing SSL tasks under varying models and scenarios (please see Line 58-76). Motivated by the identified key properties of SCUT, we focus on devising a general solution (Prompter) that can be flexibly applied to different state-of-the-art skeleton-based person re-ID models (e.g., graph transformer, GAT, Siamese encoders). This is fundamentally different from MS2L [5] and [6] that rely on a certain action recognition backbone or model (i.e., GRU or STTFormer) to design effective masking strategies. Therefore, our method could be more general and scalable than these methods. Prompter can also be flexibly applied to RGB-estimated skeletons, unsupervised scenarios, different graph modeling, and cross-domain person re-ID tasks (see Further Analyses in our paper).
>
>
> References:
>
> [1] Fan, Chao, et al. "SkeletonGait: Gait Recognition Using Skeleton Maps." Proceedings of the AAAI Conference on Artificial Intelligence, 2024.
>
> [2] Fu, Yang, et al. "Gpgait: Generalized pose-based gait recognition." Proceedings of the IEEE/CVF International Conference on Computer Vision, 2023.
>
> [3] Huang, Xiaohu, et al. "Condition-adaptive graph convolution learning for skeleton-based gait recognition." IEEE Transactions on Image Processing, 2023.
>
> [4] Wu, Wenhan, et al. "Skeletonmae: Spatial-temporal masked autoencoders for self-supervised skeleton action recognition." 2023 IEEE international conference on multimedia and expo workshops (ICMEW), IEEE, 2023.
>
> [5] Lin, Lilang, et al. "Ms2l: Multi-task self-supervised learning for skeleton based action recognition." Proceedings of the 28th ACM international conference on multimedia, 2020.
>
> [6] Zhang, Cun et al. “Spatial transformer network on skeleton‐based gait recognition.” Expert System, 2023.

---

> ### Comment · Reviewer_iRXh · 2024-11-24
> **Response to Author Rebuttal**
>
> Thank you for your detailed rebuttal, including the comprehensive clarification and experimental results.
>
> The distinctions between skeleton-based person re-identification and skeleton-based gait recognition presented by the authors are not convincing. Here are my points of contention:
>
> 1. **Focused Problem**: Gait recognition methods also involve matching and retrieval, utilizing a probe set and a gallery set during testing. I recommend that the authors review these details carefully.
>
> 2. **Input Skeleton Type**: Both 3D and 2D skeletons are applicable in skeleton-based gait recognition. For example, the Gait3D [1] and OU-MVLP-Pose [2] datasets provide 3D skeletons.
>
> 3. **Application Scenarios**: The method of acquiring skeletons, whether through sensors or pose estimators, should not impede the application of gait recognition. In fact, gait recognition can function with sensor-based skeletons.
>
> 4. **Base Architecture**: Skeleton-based gait recognition methods extend beyond GCN and CNN architectures, including transformer-based approaches [1].
>
> 5. **Datasets**: Please refer to point (2).
>
> 6. **Learning Scenarios**: This is the only aspect that holds some merit, as there are currently no self-supervised methods for skeleton-based gait recognition.
>
> 7. **Skeletal Topology**: Methods like GPGait [4] and CAG [5] employ topologies with varying semantics and levels.
>
> There is limited scope for the authors to rebut this point, as both tasks aim to identify individuals based on spatial-temporal skeleton representations, which are fundamentally similar.
>
> I appreciate the authors' additional experimental results that demonstrate the method's superiority.
>
> Regarding the masking technique, I acknowledge that the proposed method exhibits some differences from previous approaches.
>
> Overall, I have decided to increase my score to 5.
>
> [1] Zheng, Jinkai, et al. "Gait recognition in the wild with dense 3d representations and a benchmark." Proceedings of the IEEE/CVF conference on computer vision and pattern recognition. 2022.
>
> [2] An, Weizhi, et al. "Performance evaluation of model-based gait on multi-view very large population database with pose sequences." IEEE transactions on biometrics, behavior, and identity science 2.4 (2020): 421-430.
>
> [3] Zhang, Cun, et al. "Spatial transformer network on skeleton‐based gait recognition.".
>
> [4] Fu, Yang, et al. "Gpgait: Generalized pose-based gait recognition." Proceedings of the IEEE/CVF International Conference on Computer Vision. 2023.
>
> [5] Huang, Xiaohu, et al. "Condition-adaptive graph convolution learning for skeleton-based gait recognition." IEEE Transactions on Image Processing (2023).

---

> ### Author Response · Authors · 2024-11-25
>
> Thanks for your swift and detailed response. We will respond to this by two parts (Part I and Part II).
>
> Part I:
>
> We appreciate that the reviewer provided some useful insights for us to better comprehend skeleton-based gait recognition, and we have clarified the focused problem based on the details checking. However, we still hope to clarify some key questions as follows:
>
> >**(Previous Comment) Q1**. "Since skeleton-based person re-ID is fundamentally similar to skeleton-based gait recognition, the authors should compare their work with more recent gait studies in terms of method design and accuracy, such as SkeletonGait [1], GPGait [2], and CAG [3]. This comparison is crucial to assess the superiority of their approach."
>
> **A1**. We would like to clarify that the comparison table we provided is based on [1]-[3] that the reviewer mentioned in the first comment (as noted in both text and table). However, the reviewer seems to think that this comparison was claimed to be applicable to all gait recognition methods, and compares other works that may not fit into the category of these three works [1]-[3] in the new comments. We hope to emphasize that the comparison in our first response illustrates the most common case and there still exist some special cases (e.g., most skeleton-based gait recognition methods use estimated 2D skeletons while there should be very few methods using 3D skeletons). As the researchers in our area may possess more expertise in skeleton-based person re-ID than gait recognition, and the discussion of differences between skeleton-based person re-ID and gait recognition is actually beyond the scope of this work, we believe that it is more reasonable and fairer to compare these two areas using the most common case/practices.
>
> >"**Q2**. Input Skeleton Type: Both 3D and 2D skeletons are applicable in skeleton-based gait recognition. For example, the Gait3D [1] and OU-MVLP-Pose [2] datasets provide 3D skeletons."
>
> **A2**. Although both 3D and 2D skeletons can be potentially used in skeleton-based gait recognition, almost all methods in this area still employ 2D skeleton data even when the included datasets contain 3D skeletons. For example, one of the state-of-the-art methods SkeletonGait [1] uses silhouettes and heatmaps generated from only 2D skeleton data on Gait3D and GREW datasets (please check their implementation codes such as pretreatment_heatmap.py). According to the latest dataset survey [4], there are only 2 public datasets (Gait3D and GREW) that originally contain 3D skeletons in existing 24 public gait recognition datasets. These two facts further verify our comparison result that most gait recognition methods generally use 2D skeletons. We also sincerely hope to clarify the error in the comment of the reviewer, i.e., the OU-MVLP-Pose dataset only provides estimated 2D skeletons instead of 3D skeletons (please check at http://www.am.sanken.osaka-u.ac.jp/BiometricDB/GaitLPPose.html).
>
> > "**Q3**. Application Scenarios: The method of acquiring skeletons, whether through sensors or pose estimators, should not impede the application of gait recognition. In fact, gait recognition can function with sensor-based skeletons."
>
> **A3**. In [1]-[3], the input skeleton type and application scenarios are indeed skeleton data estimated from RGB videos (also called model-estimated skeletons), instead of Kinect-based/sensor-based skeletons, which is included in the comparison of our first response. After careful checking, here we hope to further clarify that this difference is also applicable to existing gait recognition methods. A crucial fact is that 24 existing gait recognition datasets either do not contain skeleton data, or ONLY contain 2D/3D skeleton data estimated from RGB videos (please check in the survey [4]). For the only two gait recognition datasets (Gait3D, GREW) that originally contain 3D pose data (Note that 3D skeletons can also be estimated from CASIA-B videos and are used in our work), Gait3D uses the 3D human mesh recovery method ROMP [5] to estimated 3D skeletons, while GREW utilizes [6] to perform this process. According to [4], none of these public gait recognition datasets have 3D skeletons that are captured from sensors (e.g., Kinect). The sensor-based skeleton data are typically more precise with less noise than estimated skeletons used in gait recognition methods due to the utilization of depth information, and they are used in almost all skeleton-based person re-ID methods. Therefore, we believe that it is also generally correct to state that the application scenarios are model-estimated skeletons for gait recognition methods.
>
> (Continued in Part II)

---

> ### Author Response · Authors · 2024-11-25
>
> Part II:
>
> >"**Q4**. Base Architecture: Skeleton-based gait recognition methods extend beyond GCN and CNN architectures, including transformer-based approaches [1]."
>
> &
>
> >"**Q5**. Skeletal Topology: Methods like GPGait [4] and CAG [5] employ topologies with varying semantics and levels."
>
> **A4** & **A5**. For base architectures, we would like to clarify that our comparison focuses on the generality of our method (e.g., can be flexibly applied to Transformer (TranSG), GAT (MG-SCR), MLP architecture (SimMC)), while [1]-[3] mainly focuses on the design of a specific GCN/CNN model (which is also the most common architecture used in gait recognition methods). For skeletal topology, GPGait [2] utilizes “a unified pose representation” (i.e., the input 2D skeletal topology is still fixed with V_in nodes and in COCO2017 format) to learn local-global relations, and CAG [3] proposes view-adaptive topology learning (VATL) to “generate view-adapative topologies for GCNs” (i.e., the input 2D skeletal topology is still fixed with N nodes). Both “varying semantics and levels” are implicitly learned in the model architectures, and they are NOT the Input skeleton topologies and levels (i.e., they do not naturally support skeleton data with different keypoint numbers). In contrast, our method flexibly supports joint-level, part-level or body-level skeleton representations with different joint numbers and body structures as inputs (please see Line 519-523 and Appendix I). So this is the key difference that we added into the comparison in our first response. We do not claim that all existing gait recognition methods possess this difference, but we aim to compare the three representative works [1]-[3] that the reviewer provided in the first comment.
>
> Based on the above clarification and the reviewer’s suggestions, we have updated the comparison table to expand its scope to all existing skeleton-based gait recognition methods (also highlighted in Appendix I). We welcome any further advice and discussion for this comparison.
>
> | Method                  |                                                                    Prompter (Ours)                                                                    |                           Skeleton-Based Gait Recognition Methods                          |
> |-------------------------|:-----------------------------------------------------------------------------------------------------------------------------------------------------:|:------------------------------------------------------------------------------------------:|
> | Task                    |                                                       3D Skeleton Based Person Re-Identification                                                      |                            2D/3D Skeleton Based Gait Recognition                           |
> | Focused Problem         |                                                   Generally Matching and Retrieving; Or Classification                                                |                                       Generally Matching and Retrieving; Or Classification                                       |
> | Input Skeleton Type     |                                                                 Generally 3D Skeletons                                                                |                                   Generally 2D Skeletons                                   |
> | Application Scenarios   |                               Generally Sensor-based skeletons;   Can be applied to Model-estimated skeletons (Explored)                              | Generally Model-estimated skeletons: Can be applied to Sensor-based skeletons (Unexplored) |
> | Base Architectures      |                                       Can be flexibly applied to different models  (e.g., Transformer, GAT, MLP)                                      |                  Generally a specific model (e.g., GCN, CNN, Transformer)                  |
> | Datasets                |                                    Generally 3D skeleton datasets (sensor-based); Datasets with estimated skeletons                                   |               22 non-skeleton datasets;  2 datasets with estimated skeletons               |
> | Learning Scenarios      |                                                          Support supervised and unsupervised                                                          |                                       Only Supervised                                      |
> | Input Skeletal Topology | Support different-level/type input skeleton data with varying nodes/topologies  (e.g., 25, 20, or 14 joints, 10 (part-level) or 5 nodes (body-level)) |      Generally unified input skeleton data with same topology (e.g., COCO2017 format)      |

---

> ### Author Response · Authors · 2024-11-25
>
> References (In both Part I and Part II):
>
> [1]Fan, Chao, et al. "SkeletonGait: Gait Recognition Using Skeleton Maps." Proceedings of the AAAI Conference on Artificial Intelligence, 2024.
>
> [2] Fu, Yang, et al. "Gpgait: Generalized pose-based gait recognition." Proceedings of the IEEE/CVF International Conference on Computer Vision, 2023.
>
> [3] Xiaohu, Huang et al. "Condition-adaptive graph convolution learning for skeleton-based gait recognition." IEEE Transactions on Image Processing, 2023.
>
> [4] Xianda, Guo, et al. “Gait Recognition in the Wild: A Large-scale Benchmark and NAS-based Baseline.” Arxiv, 2022.
>
> [5] Yu, Sun et al. “Monocular, one-stage, regression of multiple 3d people.” ICCV, 2021.
>
> [6] Ching-Hang, Chen et al. “3D human pose estimation = 2D pose estimation + matching.” CVPR, 2017.

---

> > ### Author Response · Authors · 2024-11-25
> >
> > We appreciate that the reviewer has acknowledged our efforts on the two most key concerns (i.e., our method’s superiority to representative gait recognition methods,  and key differences/novelties compared to previous masking approaches). However, as we mentioned before, the focus of our work is skeleton-based person re-ID instead of gait recognition, and the discussion of differences between skeleton-based person re-ID and gait recognition is actually beyond the scope of this work (i.e., providing it or not will not influence the technical soundness, novelty, and completeness of our work), we sincerely hope that the additional comparison offered beyond our primary focus serves only as an open technical discussion rather than a basis for evaluation. Thank you again for your detailed comments and suggestions.

---

> > > ### Comment · Reviewer_iRXh · 2024-11-25
> > > **Response to Author Feedback**
> > >
> > > Thank you to the authors for providing the additional clarifications.
> > >
> > > My primary concern with this work is that the 3D skeleton-based re-identification (re-ID) and skeleton-based gait recognition studied in this paper are essentially the same task. Both aim to learn the spatiotemporal features of the skeleton to identify a person’s identity. Using different datasets to validate these two tasks does not change the fact that they are fundamentally the same. Moreover, skeleton-based gait recognition has been developed earlier and more extensively than 3D skeleton-based re-ID. I believe these two research directions should not be separated. Developing two essentially identical tasks into two sets of evaluation systems is not beneficial for the advancement of related research. I hope we can conduct research from a more unified perspective and avoid writing papers solely for the sake of publication (just as I would not reject a paper simply for the sake of rejecting it).
> > >
> > > I appreciate the authors’ time and effort during the discussion period, but I will maintain my score. Regarding the discussions during the rebuttal period, I would like to leave the final decision on this matter to the area chair.
> > >
> > > Thanks.

---

> ### Author Response · Authors · 2024-11-28
> **Thank You for Your Valuable Feedback  & Further Response - Part I**
>
> Thank you for your valuable feedback and discussions again.
>
> We admire the insistence of the reviewer to unify two research directions, 3D skeleton based person re-ID and skeleton-based gait recognition, to advance the skeleton-based AI innovations in both communities. Nevertheless, we still hope to classify the essential differences between these two directions from a more holistic perspective:
>
> 1. **3D Skeleton Based Person Re-ID Methods** (The focused area) VS. **Earliest and Most Skeleton-Based Gait Recognition Methods**
>
>     - Although the first pose-based gait recognition (i.e., skeleton-based gait recognition) approach [1] was proposed in 2017 (according to [2]), it is like most existing skeleton-based gait recognition methods (see the response below) that utilizes **2D estimated skeleton data**, which differs from the 3D skeleton based person re-ID methods.
>
>     >”(2) Although both 3D and 2D skeletons can be potentially used in skeleton-based gait recognition, almost all methods in this area still employ 2D skeleton data even when the included datasets contain 3D skeletons. For example, one of the state-of-the-art methods SkeletonGait [1] uses silhouettes and heatmaps generated from only 2D skeleton data on Gait3D and GREW datasets (please check their implementation codes such as pretreatment_heatmap.py). According to the latest dataset survey [4], there are only 2 public datasets (Gait3D and GREW) that originally contain 3D skeletons in existing 24 public gait recognition datasets. These two facts further verify our comparison result that most gait recognition methods generally use 2D skeletons. We also sincerely hope to clarify the error in the comment of the reviewer, i.e., the OU-MVLP-Pose dataset only provides estimated 2D skeletons instead of 3D skeletons (please check at http://www.am.sanken.osaka-u.ac.jp/BiometricDB/GaitLPPose.html).”
>
>     - For the technical difference between 2D and 3D skeletons (can be viewed as 2D and 3D signals), we hope to reference [3] and the opinions in [4]-[5] from Dr. Feifei Li and her team to better illustrate (the reviewer can optionally refer to these talks about “3D spatial intelligence” if interested). The key point is that although 2D signals like 2D skeletons estimated from RGB images can be transformed to help us and AI models perceive 3D world (e.g., human brains can receive 2D visual signals to intricately transform them into the 3D sensing), they are NOT like the 3D signals that naturally follow the laws of physics and own inherent 3D structures and attributes of the real world (e.g., the 3D skeleton data captured from Kinect possess physical depth information and structure). In this context, “3D representations could typically produce a better fit than 2D representations between models and real-world tasks” -- Dr. Feifei Li. In many public invited talks by our team, we have shared this opinion to various researchers and most of them agree with it. This is also the primary motivation for us to utilize 3D skeleton data captured from depth sensors, and partially explains why we think 3D skeleton based person re-ID methods are different from the earliest and most skeleton based gait recognition methods that generally utilize 2D skeletons (detailed in the comparison table we updated in the previous response).
>
> 2. **3D Skeleton Based Person Re-ID Methods** (The focused area) VS. **Earliest 3D Skeleton Based Gait Recognition Methods**
>
>     - To the best of our knowledge, the earliest 3D skeleton based person re-identification method [8] and the first 3D skeleton based gait recognition method [6] were proposed in the **same year** (2020) [2][7]. Based on this fact, we think it is hard to assert that they belong to the same branch, as none of these works actually inherits from each other (despite some similarity such as gait related), just like that we cannot directly unify the person re-ID area (concept first appears in 2005 [9]) and gait recognition area (concept first appears in 1996 [7]). Both these two different research communities are large and have independent and unique development in the past tens of years. Moreover, I think both 3D skeleton based person re-ID methods (over 10 impactful works and all are compared in our work) and 3D skeleton based gait recognition methods have been extensively developed in the past few years. We highly value the unified perspective of the reviewer, but considering nearly identical timeline of emergence, inherent differences (detailed in the comparison table we updated in the previous response), rapid and independent development of both two research communities, and their respective high impact, we believe that these two research branches should be separated but they can be better developed from diverse insights of each other (just like this insightful discussion between the reviewer and us).

---

> ### Author Response · Authors · 2024-11-28
> **Thank You for Your Valuable Feedback  & Further Response - Part II**
>
> - Moreover, we understand that the reviewer may expect that the proposed general method can be applied to not only skeleton-based person re-ID but also other areas (e.g., gait recognition, action recognition) to advance the development of different related research. To this end, we have additionally integrated the proposed Prompter method into the representative state-of-the-art skeleton-based gait recognition method (GPGait) and skeleton-based action recognition method (ST-GCN) to verify the effectiveness and generality of our method. As shown in the table below (also highlighted in Appendix I, Sec. E.5), our method can simultaneously improve the performance (Rank-1 accuracy) of both gait recognition and action recognition methods in most cases on different datasets, which further demonstrates the general applicability of our method and suggests its great potential for more related areas.
>
> | Method Source (Research Community) | Method            | KS20 | IAS-A | IAS-B | KGBD |
> |------------------------------------|-------------------|:----:|:-----:|:-----:|:----:|
> |  Person Re-Identification          | TranSG [12]        | 71.3 |  48.0 |  56.1 | 57.0 |
> |                                    | + Prompter (Ours) | **74.2** |  **49.5** |  **60.4** | **59.5** |
> |  Gait Recognition                  | GPGait [10]        | 71.4 |  50.9 |  60.1 | 53.6 |
> |                                    | + Prompter (Ours) | **72.7** |  **55.3** |  **61.7** | 53.4 |
> |  Action Recognition                | ST-GCN [11]        | 60.4 |  43.6 |  49.1 | 57.7 |
> |                                    | + Prompter (Ours) | **65.6** |  **53.4** |  **58.8** | **59.0** |
>
>
> **We are heartfelt grateful for the detailed comments and open insightful discussions with the reviewer. We sincerely hope the reviewer and the area chair can take our clarifications into account.**
>
> Best Regards.
>
> The Authors
>
> References:
>
> [1] Liao et al. “Pose-based temporal-spatial network (PTSN) for gait recognition with carrying and clothing variations.” Biometric Recognition, 2017.
>
> [2] Teepe et al. “Towards a Deeper Understanding of Skeleton-based Gait Recognition.” CVPR workshop, 2022.
>
> [3] Han et al. “Space-time representation of people based on 3D skeletal data: A review.” Computer Vision and Image Understanding, 2017.
>
> [4] Li et al. “With Spatial Intelligence, AI Will Understand the Real World.” TED Talk, 2024.
>
> [5] Li et al. “The Future of AI is Here — Fei-Fei Li Unveils the Next Frontier of AI.” a16z Interview, 2024.
>
> [6] Liao et al. “A model-based gait recognition method with body pose and human prior knowledge.” Pattern Recognition, 2020.
>
> [7] Shen et al. “A Comprehensive Survey on Deep Gait Recognition: Algorithms, Datasets, and Challenges.” IEEE Transactions on Biometrics, Behavior, and Identity Science, 2024.
>
> [8] Rao et al. “Self-supervised gait encoding with locality-aware attention for person re-identification.” IJCAI, 2020.
>
> [9] Zheng et al. “Person Re-identification: Past, Present and Future.” 2016.
>
> [10] Yang et al. "Gpgait: Generalized pose-based gait recognition." ICCV, 2023.
>
> [11] Yan et al. "Spatial temporal graph convolutional networks for skeleton-based action recognition." AAAI, 2018.
>
> [12] Rao et al. "TranSG: Transformer-Based Skeleton Graph Prototype Contrastive Learning with Structure-Trajectory Prompted Reconstruction for Person Re-Identification." CVPR, 2023.

---

> ### Author Response · Authors · 2024-11-30
> **Thanks to Reviewer iRXh**
>
> Dear Reviewer iRXh,
>
> We would like to thank you again for your valuable feedback to improve our work. We are wondering whether our response has addressed your questions and can improve your opinion of our work.
>
> Kindly let us know if you might have further comments, and we will do our best to address them.
>
> Best Regards,
>
> The Authors

---

> > ### Comment · Reviewer_iRXh · 2024-12-01
> >
> > Dear Authors,
> >
> > Thank you for your response. I am encouraged by your provision of additional results on gait and action recognition methods.
> >
> > > **(1) 3D Skeleton-Based Person Re-ID Methods (Focused Area) vs. Earliest and Most Skeleton-Based Gait Recognition Methods**
> >
> > Regarding this point, I recognize the differences between 3D and 2D skeletons, as they are derived through different approaches and have distinct physical representations. However, this does not substantiate a fundamental difference between skeleton-based gait recognition and re-ID.
> >
> > The reasons are twofold:
> >
> > a. Conceptually, the two tasks are identical, as both aim to recognize individuals by modeling skeleton features.
> >
> > b. The methods used to model 3D or 2D skeletons (MLPs/Transformers/GCNs, excluding heatmap-based methods) exhibit negligible differences. By simply modifying the input channels to either 3 or 2, all methods can be applied to both 3D and 2D skeletons.
> >
> > Therefore, from both conceptual and methodological perspectives, it is challenging to differentiate between 3D skeleton-based person re-ID and gait recognition methods.
> >
> > > **(2) 3D Skeleton-Based Person Re-ID Methods (Focused Area) vs. Earliest 3D Skeleton-Based Gait Recognition Methods**
> >
> > The authors mention that "just like that we cannot directly unify the person re-ID area (a concept first appearing in 2005 [9]) and the gait recognition area." I agree that **appearance-based** person re-ID and gait recognition cannot be unified, as they utilize different modalities—natural images and silhouettes—and therefore engage different methodological branches. However, skeleton-based person re-ID and gait recognition share the same modalities, allowing the modeling methods to be readily applicable to both.
> >
> > Furthermore, as noted by Reviewer DvW6, all recent methods from 2021 compared in Table 2 originate from the same group of authors. This suggests that this research direction might not be as influential for a broader audience.
> >
> > Considering all these points, I maintain my current score, which is my final decision.
> >
> > Thank you.

---

> ### Author Response · Authors · 2024-12-01
> **Thank You for Your Valuable Comment & Further Response - Part I**
>
> Dear Reviewer,
>
> We appreciate your valuable feedback and your recognition/agreement with some of our responses. We would like to further answer your questions below.
>
> >**Q1**. ”a. Conceptually, the two tasks are identical, as both aim to recognize individuals by modeling skeleton features.”
>
> **A1**. Thank you for your valuable comment. We would like to point out that **having a similar goal does NOT mean that these two areas are identical** (please see the detailed comparison between our method and skeleton-based gait recognition methods in the previous response and below). A good example is that both conventional person re-ID methods and gait recognition methods aim to recognize individuals by modeling human body features (e.g., appearance-based RGB or silhouette features), they are still regarded as **conceptually and traditionally distinct fields**.
>
>
> >>| Method                  |                                                                    Prompter (Ours)                                                                    |                           Skeleton-Based Gait Recognition Methods                          |
> |-------------------------|:-----------------------------------------------------------------------------------------------------------------------------------------------------:|:------------------------------------------------------------------------------------------:|
> | Task                    |                                                       3D Skeleton Based Person Re-Identification                                                      |                            2D/3D Skeleton Based Gait Recognition                           |
> | Focused Problem         |                                                   Generally Matching and Retrieving; Or Classification                                                |                                       Generally Matching and Retrieving; Or Classification                                       |
> | Input Skeleton Type     |                                                                 Generally 3D Skeletons                                                                |                                   Generally 2D Skeletons                                   |
> | Application Scenarios   |                               Generally Sensor-based skeletons;   Can be applied to Model-estimated skeletons (Explored)                              | Generally Model-estimated skeletons: Can be applied to Sensor-based skeletons (Unexplored) |
> | Base Architectures      |                                       Can be flexibly applied to different models  (e.g., Transformer, GAT, MLP)                                      |                  Generally a specific model (e.g., GCN, CNN, Transformer)                  |
> | Datasets                |                                    Generally 3D skeleton datasets (sensor-based); Datasets with estimated skeletons                                   |               22 non-skeleton datasets;  2 datasets with estimated skeletons               |
> | Learning Scenarios      |                                                          Support supervised and unsupervised                                                          |                                       Only Supervised                                      |
> | Input Skeletal Topology | Support different-level/type input skeleton data with varying nodes/topologies  (e.g., 25, 20, or 14 joints, 10 (part-level) or 5 nodes (body-level)) |      Generally unified input skeleton data with same topology (e.g., COCO2017 format)      |

---

> ### Author Response · Authors · 2024-12-01
> **Thank You for Your Valuable Comment & Further Response - Part II**
>
> >**Q2**. ”b. The methods used to model 3D or 2D skeletons (MLPs/Transformers/GCNs, excluding heatmap-based methods) exhibit negligible differences. By simply modifying the input channels to either 3 or 2, all methods can be applied to both 3D and 2D skeletons.”
>
> **A2**. Thank you for your valuable response. We hope to clarify that **simply modifying the input channels may NOT ensure the good adaptability, generality or effectiveness of the model on a new modality (e.g., 2D signals to 3D signals), due to the inherent difference between 2D and 3D skeletons as the reviewer acknowledged**. We provide a comparison as an example below, which shows that although the methods, which originally use estimated 2D skeletons or are specifically designed for gait recognition, can be applied to 3D skeletons, they might **perform much more poorly than the used baseline method** TranSG (here we use the baseline method to better highlight the difference) specifically designed for 3D skeleton based person re-ID under the same evaluation setting on different datasets. These results suggest that the skeleton-based gait recognition methods (regardless of the state-of-the-art SkeletonGait or classic PoseGait) **should be further designed or modified** in terms of both architectures (e.g., simpler but more effective architectures), learning mechanisms, or feature representations to be adapted to 3D skeleton based person re-ID task. This could provide additional insights and proof for the area gap between 3D skeleton-based person re-ID and gait recognition methods.
>
> |  Method         Type                            |       Method                  | KS20 |        | BIWI-W |        | BIWI-S |        | IAS-A |        | IAS-B |        | KGBD |        |
> |---------------------------------------------|-------------------------|:----:|--------|:------:|--------|:------:|:------:|:-----:|--------|:-----:|--------|:----:|--------|
> |                                             |                         |  mAP | Rank-1 |    mAP | Rank-1 |    mAP | Rank-1 |   mAP | Rank-1 |   mAP | Rank-1 |  mAP | Rank-1 |
> | 2D Skeleton Based Gait Recognition Method  | SkeletonGait (Skeleton) | 14.5 |  22.2  |  11.2  |  10.8  |   9.3  |  15.1  |  25.0 |  31.4  |  21.5 |  31.5  |  1.1 |   1.7  |
> | 3D Skeleton Based Gait Recognition Method  | PoseGait                | 23.5 |  49.4  |  11.1  |   8.8  |   9.9  |  14.0  |  17.5 |  28.4  |  20.8 |  28.9  | 13.9 |  50.6  |
> | 3D Skeleton Based Person Re-ID Method      | TranSG (Our Baseline)   | 42.5 |  71.3  |  25.5  |  31.2  |  26.7  |  66.6  |  31.8 |  48.0  |  37.9 |  56.1  | 18.1 |  57.0  |
>
> >>” We highly value the unified perspective of the reviewer, but considering nearly identical timeline of emergence, inherent differences (detailed in the comparison table we updated in the previous response), rapid and independent development of both two research communities, and their respective high impact, we believe that these two research branches should be separated but they can be better developed from diverse insights of each other (just like this insightful discussion between the reviewer and us).”
>
> **Based on the the provided qualitative and empirical comparison above and other many factors below (please see previous response above), we think that we need to differentiate between 3D skeleton-based person re-ID and gait recognition methods.**
>
>
> >**Q3**. ”I agree that appearance-based person re-ID and gait recognition cannot be unified, as they utilize different modalities—natural images and silhouettes—and therefore engage different methodological branches. However, skeleton-based person re-ID and gait recognition share the same modalities, allowing the modeling methods to be readily applicable to both.”
>
> **A3**. Thank you for your valuable feedback. We appreciate that the reviewer acknowledged that appearance-based person re-ID and gait recognition cannot be unified, and this opinion/principle **should ALSO be applicable to skeleton-based person re-ID and gait recognition**. In fact, the appearance-based person re-ID methods also widely employ silhouettes [1]-[7], while appearance-based gait recognition methods can utilize not only silhouettes but also RGB images [8]. Therefore, they essentially share similar modalities (e.g., silhouettes can actually be viewed as appearance-based features and can be extracted from RGB images/videos) but this does NOT change the fact that they belong to two separate areas, which also holds true for skeleton-based person re-ID and gait recognition.

---

> ### Author Response · Authors · 2024-12-01
> **Thank You for Your Valuable Comment & Further Response - Part III**
>
> >**Q4**. ”Furthermore, as noted by Reviewer DvW6, all recent methods from 2021 compared in Table 2 originate from the same group of authors. This suggests that this research direction might not be as influential for a broader audience.”
>
> **A4**. Thank you for your valuable comment.
> - Firstly, we would like to point out that **our method is compared with all existing state-of-the-art skeleton-based person re-ID methods, regardless of whether they belong to the same or different groups** (i.e., methods from the same or different groups will not influence our selection and comparison criteria based on the sufficient technical soundness, novelty, and impact of these methods). All methods are fairly compared using the same evaluation protocol.
>
> - Secondly, **our method is actually compared with 20 approaches from diverse groups of authors (up to 13 different groups)**, and here we provide an overview of compared methods (we present the most representative method from the same group and also include newly added during the revision) in our work for both reviewers and area chair:
>
> | Comparison Part                                    | Method  Type                          | Method       | Source Group                          |
> |----------------------------------------------------|---------------------------------------|--------------|---------------------------------------|
> | Sec. 4.2 & Revision (Added to Appendices or Paper) | Skeleton-Based Person Re-ID Method    | D_PG         | Liao et al., 2020 (Totally 2 Methods) |
> |                                                    |                                       | D_13         | Munaro et al., 2014                   |
> |                                                    |                                       | D_16         | Pala et al., 2019                     |
> |                                                    |                                       | TranSG       | Rao et al., 2023 (Totally 7 Methods)  |
> |                                                    | Skeleton-Based Gait Recognition Method | SkeletonGait | Chao et al., 2024                     |
> |                                                    |                                       | GaitTR       | Cun et al., 2023                      |
> |                                                    |                                       | GPGait       | Yang et al., 2023                     |
> |                                                    | Action Recognition Method             | ST-GCN       | Yan et al., 2018                      |
> | Sec. 4.3                                           | Person Re-ID Method                   | LMNN         | Weinberger & Saul, 2009               |
> |                                                    |                                       | ITML         | Davis et al., 2007                    |
> |                                                    |                                       | ELF          | Gray & Tao, 2008                      |
> |                                                    |                                       | SDALF        | Farenzena et al., 2010                |
> |                                                    |                                       | MLR          | Liu et al., 2015                      |
>
> - **We acknowledge that seven state-of-the-art skeleton-based person re-ID methods (*35% of all methods*) originate from the same group**, which developed **the first skeleton-based person re-ID method as early as 2020 [9]** and has pioneered numerous skeleton-based innovations in both person re-ID and action recognition communities [10]. It is worth noting that most existing skeleton-based person re-ID works also follow the settings and designs from this research group. However, with a systematic evaluation of the skeleton semantics learning (SSL) tasks used in these methods based on the proposed SCUT framework, we **identify the bottlenecks in their generality and performance**, including the lack and limit of co-training compatibility (CTC) and spatial-temporal effectiveness (STE) (please see Line 70-76, Table 1 and Line 367-377 of our paper), which negatively influence their performance when applied to different models and scenarios (e.g., different types of skeleton data) (please see Table 3, 5 and Sec. 4.3 of our paper). This motivates us to propose the general SSL task Prompter to address these challenges to facilitate the performance of different state-of-the-art skeleton-based person re-ID models under different scenarios (please see Line Sec. 4.2, Table 2 of our paper).

---

> ### Author Response · Authors · 2024-12-01
> **Thank You for Your Valuable Comment & Further Response - Part IV**
>
> - Moreover, **our work is also the first exploration and assessment of the multi-faceted generality of existing SSL tasks under different scenarios, and presents the first systematic generality assessment framework** termed SCUT that identifies and quantifies key characteristics of SSL tasks in terms of Spatial-temporal effectiveness (STE), Co-training compatibility (CTC), Unsupervised trainability (UT), and Task transformability (TT) (please see the summary of novelty and generality of our work in the response to Reviewer BHkC). Both the proposed SSL generality assessment framework and the proposed Prompter method could provide valuable insights and have the potential to advance broader related areas (as recognized by different reviewers such as v2zj and DUn6, please see the General Response):
>
>
> References:
>
> [1] Nambiar et al. Gait-based person re-identification: A survey. ACM Computing Surveys, 2019
>
> [2] Liu et al. Enhancing person re-identification by integrating gait biometric. Neurocomputing, 2015.
>
> [3] Gala et al. Gait-assisted person re-identification in wide area surveillance. ACCV Workshop, 2014.
>
> [4] Wei et al. Swiss-system based cascade ranking for gait-based person re-identification. AAAI, 2015
>
> [5] Kawei et al. Person re-identification using view-dependent score-level fusion of gait and color features. ICPR, 2012
>
> [6] Roy et al. Hierarchical method combining gait and phase of motion with spatiotemporal model for person re-identification. Pattern Recognition, 2012.
>
> [7] Bai et al. Incorporating texture and silhouette for video-based person re-identification. Pattern Recognition, 2024.
>
> [8] Şahan et al. A survey of appearance-based approaches for human gait recognition: techniques, challenges, and future directions. The Journal of Supercomputing, 2024.
>
> [9] Rao et al. “Self-supervised gait encoding with locality-aware attention for person re-identification.” IJCAI, 2020.
>
> [10] Rao et al. “Augmented Skeleton Based Contrastive Action Learning with Momentum LSTM for Unsupervised Action Recognition” Information Sciences, 2021.
>
>  - We deeply appreciate the reviewer's detailed feedback and valuable comments. In our future work, we will include more up-to-date methods from various related areas (e.g., gait recognition) for a more comprehensive comparison. Furthermore, **all discussions regarding the differences and similarities between skeleton-based person re-ID and gait recognition will be incorporated into the future version of our paper to provide readers with a clearer understanding of these two areas**.
>
> We sincerely hope reviewers and the area chair can take our above clarifications into consideration.
>
> Best Regards,
>
> The Authors

---

### Official Review · Reviewer_DvW6 · 2024-11-03

**Soundness:** 4
**Presentation:** 3
**Contribution:** 2
**Rating:** 5
**Confidence:** 3

**Summary:**

This paper addresses person re-identification (re-ID) via skeleton data, aiming to improve generalizability in safety-critical applications. The authors introduce SCUT, an SSL generality assessment framework that identifies four key properties: Spatial-temporal effectiveness, Co-training compatibility, Unsupervised trainability, and Task transformability.  Furthermore, they introduced Prompter task to probabilistically masks joint positions and motion to capture robust spatial-temporal patterns.

**Strengths:**

1.This paper is well organized. Writing is clear and easy to follow.
2.The experiments are extensive and thorough, providing convincing evidence of the proposed method's effectiveness. The algorithm consistently outperforms baselines across various settings, demonstrating its potential and robustness in skeleton-based person re-ID.

**Weaknesses:**

The motivation for reconstructing individual joint locations or trajectories in joint-level is unclear. While part-level or body-level dynamics provide intuitive insights into human movement, joint-level information can be noisy. In datasets like BIWI-W, where the state-of-the-art Rank-1 accuracy is only 34.6% across 50 individuals, human poses can be ambiguous, suggesting limitations in relying on fine-grained joint-level details for re-ID tasks.

**Questions:**

1. Could you clarify the motivation for using Prompter? Specifically, is there empirical evidence that compares the effectiveness of joint-level reconstruction against part-level or body-level reconstruction approaches?
2. Are there any qualitative examples or insights into how Prompter’s performance generalizes across datasets with different characteristics or collection methods? For instance, while IAS-A and IAS-B achieve MAP scores of 34.1 and 43.8 and Rank-1 scores of 39.5 and 60.4, respectively, when trained individually, these metrics decrease to MAP scores of 19.1 and 18.5 and Rank-1 scores of 35.8 and 34.9 when transferred from BIWI. Could you clarify which features Prompter learns that transfer well across datasets and which are specifically learned within each dataset?

---

> ### Author Response · Authors · 2024-11-24
>
> Thank you for taking the time to review our paper and providing valuable feedback. We would like to answer your questions below.
>
> >**Q1**. The motivation for reconstructing individual joint locations or trajectories in joint-level is unclear. While part-level or body-level dynamics provide intuitive insights into human movement, joint-level information can be noisy. In datasets like BIWI-W, where the state-of-the-art Rank-1 accuracy is only 34.6% across 50 individuals, human poses can be ambiguous, suggesting limitations in relying on fine-grained joint-level details for re-ID tasks.
>
> &
>
> >**Q2**. Could you clarify the motivation for using Prompter? Specifically, is there empirical evidence that compares the effectiveness of joint-level reconstruction against part-level or body-level reconstruction approaches?
>
> **A1** & **A2**. Thank you for your valuable question and suggestion.
>
> - Firstly, we have provided additional experiments to evaluate our method on part-level and body-level skeleton representations following previous skeleton-based person re-ID methods (please see Line 201-204 in Appendix I), and compare its performance with other state-of-the-art skeleton semantics learning (SSL) task (DR, MIC, STPR) following our main experiments. As presented in the table below (also provided in Appendix I Sec. E.2 Table 11), the results show that our method outperforms different state-of-the-art SSL tasks on both original and higher-level skeleton representations in most cases (highlighted in Line 521-523 in our paper). This demonstrates its generality and stronger effectiveness under different-level skeletal structures to facilitate the model to learn more discriminative features.
>
> - Secondly, it is also observed that using joint-level reconstruction achieves better performance than both part-level and body-level reconstruction on different datasets, regardless of using different reconstruction mechanisms (DR, STPR) and SSL tasks (MIC). This implies the part-level and body-level skeleton representations may contain less specific positional and motion information than the original joint-level representations, and their more abstract part construction might negatively influence performance of different semantics learning tasks. These results and observations are also consistent with previous works [3].
>
> - Moreover, we would like to point out that the 3D skeleton data (i.e., 3D body joint coordinates) in four public benchmark datasets (KGBD, KS20, BIWI, IAS) are captured from depth sensors (i.e., Kinect), which physically detect depth information to generate skeletons that typically possess higher precision and lower noise [1]-[2] than 2D/3D skeleton data extracted from RGB videos using pose estimation models. Therefore, they can be viewed as ground-truth skeleton data to accurately reflect human movement for effective skeleton pattern learning. Base on this reason and considering the higher effectiveness of joint-level information in practice (please see (1)-(2)), we follow previous works to mainly use joint-level information in terms of structural locations and motion trajectories (please see motivation in Line 290-306, 377-401) for skeleton semantics learning and person re-ID, while we also discuss performance of different-level skeleton representations in our paper (please see Line 519-523, Sec. 4.3 and Appendix I).
>
>
> |                 | BIWI-W      |            |            | KS20        |            |            |
> |-----------------|-------------|------------|------------|-------------|------------|------------|
> |                 | Joint-Level | Part-Level | Body-Level | Joint-Level | Part-Level | Body-Level |
> | DR              | 33.8        | 17.6       | 13.2       |     73.2    |    48.4    |    39.3    |
> | MIC             | 34.5        | 19.1       | 12.2       |     72.3    |    48.4    |    40.8    |
> | STPR            | 32.7        | 20.0       | 17.0       |     73.6    |    48.1    |    40.8    |
> | Prompter (Ours) | **34.6**        | **20.1**       | 16.3       |     **74.2**    |    **49.4**    |    **41.9**    |
>
> References:
>
> [1] Han et al. Space-time representation of people based on 3D skeletal data: A review. Computer Vision and Image Understanding, 2017.
>
> [2] Rao et al. A self-supervised gait encoding approach with locality-awareness for 3D skeleton based person re-identification. IEEE Transactions on Pattern Analysis and Machine Intelligence, 2021.
>
> [3] Rao et al. Hierarchical skeleton meta-prototype contrastive learning with hard skeleton mining for unsupervised person re-identification. International Journal of Computer Vision, 2021.

---

> ### Author Response · Authors · 2024-11-24
>
> >**Q3**. Are there any qualitative examples or insights into how Prompter’s performance generalizes across datasets with different characteristics or collection methods? For instance, while IAS-A and IAS-B achieve MAP scores of 34.1 and 43.8 and Rank-1 scores of 39.5 and 60.4, respectively, when trained individually, these metrics decrease to MAP scores of 19.1 and 18.5 and Rank-1 scores of 35.8 and 34.9 when transferred from BIWI. Could you clarify which features Prompter learns that transfer well across datasets and which are specifically learned within each dataset?
>
> **A3**. Thanks for your constructive suggestions. We have provided additional qualitative analyses for the generalized performance across datasets, including confusing matrices for analyzing accuracy differences on different classes, and t-SNE feature visualization for analyzing the intra-class and inter-class similarity among classes (please see Appendix I Sec. E.2, Fig. 11, 12, 13): (1) Firstly, as shown in Fig. 11 of Appendix I, when generalizing the model pre-trained on BIWI to IAS testing sets, the accuracy on class 4 and class 9 evidently decreases on IAS-A (which might suggest a domain shift between different datasets and is consistent with the observations in [1]), while the wrong predictions between class 1 and class 2 also decrease, which implies that the features learned on BIWI may help the model to better distinguish them. Similar results are observed on IAS-B, and we can intuitively analyze the generalized effectiveness of features trained on other datasets (e.g., BIWI) by comparing accuracies in the provided confusion matrices. (2) Secondly, in the results of feature visualization (please see Fig. 12 of Appendix I), we can see that the model trained on BIWI can generalize to IAS training set with relatively distinct class separation especially on class 0, class 1, class 8, class 10, which demonstrates the effectiveness of our method to general skeleton semantics that can be applied to different datasets without model fine-tuning. However, due to the inherent domain shifts (i.e., domain distribution differences) between different datasets (e.g., BIWI and IAS), we can see that many class features (e.g., class 5) that are hard to be learned on the original datasets still cannot be well classified in the cross-domain person re-ID task, which is also consistent with the results in confusion matrices in Fig. 11. The qualitative results in both confusion matrices and t-SNE feature visualization provide an intuitive explanation for the performance changes in the cross-domain person re-ID task.
>
>
> References:
>
> [1] Rao et al. A self-supervised gait encoding approach with locality-awareness for 3D skeleton based person re-identification. IEEE Transactions on Pattern Analysis and Machine Intelligence, 2021.
>
> We sincerely hope our clarifications above have addressed your concerns and can improve your opinion of our work.

---

> ### Comment · Reviewer_DvW6 · 2024-11-27
>
> I appreciate the authors' efforts in conducting empirical experiments.
>
> The qualitative analyses for the generalized performance across datasets authors provided that address my concern.
>
> However, I agree with Reviewer BHkC that the theoretical contributions underlying the overall masking and reconstruction approach in Prompter are relatively limited. Additionally, the choice of joint-level reconstruction appears to be driven primarily by empirical observations rather than theoretical justification.
>
> I also have concerns regarding the potential impact of the Skeleton Semantics Learning (SSL) task on the broader machine learning community. Both the related work section and the baselines primarily reference literature from a single group of authors. Additionally, SSL presented in the paper is focused solely on human skeletons, the  it appears to overlap with existing areas such as skeleton-based gait and action recognition. I believe the authors could further distinguish the SSL task by extending the analysis of skeletons to rigid, affine, or articulated objects, which might help demonstrate the unique contribution of SSL and its broader applicability. I’ll leave the final decision of this matter to AC.
>
> I have raised the soundness score to excellent; however, the overall score remains unchanged due to the remaining concern.

---

> ### Author Response · Authors · 2024-11-28
> **Thank You for Raising Soundness Score & Your Valuable Feedback & Further Response - Part I**
>
> We are glad to know that our response has addressed your previous questions, and sincerely thank you for improving the soundness score. We would like to answer your new questions below.
>
> >**Q1**. ” the theoretical contributions underlying the overall masking and reconstruction approach in Prompter are relatively limited. Additionally, the choice of joint-level reconstruction appears to be driven primarily by empirical observations rather than theoretical justification.”
>
> **A1**. Thank you for your valuable feedback.
> - For the technical novelty and contributions, we have further clarified and detailed in the response to the Reviwer BHkC (please see our responses to BHkC). We would like to further clafiry that we not only provide comprehensive empiical/quantitative evaluations of the proposed method (please see Sec. 4.2, 4.3 and Appendix), but also have offered extensive qualitative analyses of its effectiveness, including:
>     1. **loss visualization** for analyzing effectiveness of Task Transformability (TT) (please see Fig. 3 of our paper), downstream loss, skeleton semantics learning loss, mean intra-class tightness (mACT), and mean inter-class looseness (mRCL) (please see Sec. C.1 of Appendix I);
>     2. **skeleton feature visualization** (please see Sec. C.2 of Appendix I) to analyze the intra-class simiarlity, inter-class difference, and method effectivenss for cross-domain person re-ID (please see Sec. E.2 of Appendix I);
>     3. **confusion matrices** (please see Sec. C.3 of Appendix I) to analyze the accuracy on different classes and the easily-confused identities, so as to provide valuable insights for other researcher to better develop the proposed method by focusing on the hard samples.
>
> - Moreover, we also have provided **theoretical** assumptions and analyses of our proposed method by innovatively modeling it as a model regularization method (please see Line 367-377 of our paper and Appendix II), and revealed its relations to existing model regularization methods like Dropout. We appreciate the valuable suggestion of the reviewer and will further provide more detailed theoretical analysis in the future version of our work.

---

> ### Author Response · Authors · 2024-11-28
> **Thank You for Raising Soundness Score & Your Valuable Feedback & Further Response - Part II (Swap Order with Part III)**
>
> >**Q2**. ”Both the related work section and the baselines primarily reference literature from a single group of authors.”
>
> **A2**. Thank you for your valuable comment.
> - Firstly, we would like to point out that **our method is compared with all existing state-of-the-art skeleton-based person re-ID methods, regardless of whether they belong to the same or different groups** (i.e., methods from the same or different groups will not influence our selection and comparison criteria based on the sufficient technical soundness, novelty, and impact of these methods). All methods are fairly compared using the same evaluation protocol.
>
> - Secondly, **our method is actually compared with 20 approaches from diverse groups of authors (up to 13 different groups)**, and here we provide an overview of compared methods (we present the most representative method from the same group and also include newly added during the revision) in our work for both reviewers and area chair:
>
> | Comparison Part                                    | Method  Type                          | Method       | Source Group                          |
> |----------------------------------------------------|---------------------------------------|--------------|---------------------------------------|
> | Sec. 4.2 & Revision (Added to Appendices or Paper) | Skeleton-Based Person Re-ID Method    | D_PG         | Liao et al., 2020 (Totally 2 Methods) |
> |                                                    |                                       | D_13         | Munaro et al., 2014                   |
> |                                                    |                                       | D_16         | Pala et al., 2019                     |
> |                                                    |                                       | TranSG       | Rao et al., 2023 (Totally 7 Methods)  |
> |                                                    | Skeleton-Based Gait Recognition Method | SkeletonGait | Chao et al., 2024                     |
> |                                                    |                                       | GaitTR       | Cun et al., 2023                      |
> |                                                    |                                       | GPGait       | Yang et al., 2023                     |
> |                                                    | Action Recognition Method             | ST-GCN       | Yan et al., 2018                      |
> | Sec. 4.3                                           | Person Re-ID Method                   | LMNN         | Weinberger & Saul, 2009               |
> |                                                    |                                       | ITML         | Davis et al., 2007                    |
> |                                                    |                                       | ELF          | Gray & Tao, 2008                      |
> |                                                    |                                       | SDALF        | Farenzena et al., 2010                |
> |                                                    |                                       | MLR          | Liu et al., 2015                      |
>
> - **We acknowledge that seven state-of-the-art skeleton-based person re-ID methods (*35% of all methods*) originate from the same group**, which developed **the first skeleton-based person re-ID method as early as 2020 [9]** and has pioneered numerous skeleton-based innovations in both person re-ID and action recognition communities [10]. It is worth noting that most existing skeleton-based person re-ID works also follow the settings and designs from this research group. However, with a systematic evaluation of the skeleton semantics learning (SSL) tasks used in these methods based on the proposed SCUT framework, we **identify the bottlenecks in their generality and performance**, including the lack and limit of co-training compatibility (CTC) and spatial-temporal effectiveness (STE) (please see Line 70-76, Table 1 and Line 367-377 of our paper), which negatively influence their performance when applied to different models and scenarios (e.g., different types of skeleton data) (please see Table 3, 5 and Sec. 4.3 of our paper). This motivates us to propose the general SSL task Prompter to address these challenges to facilitate the performance of different state-of-the-art skeleton-based person re-ID models under different scenarios (please see Sec. 4.2, Table 2 of our paper).

---

> ### Author Response · Authors · 2024-11-30
> **Thanks to Reviewer DvW6**
>
> Dear Reviewer DvW6,
>
> We would like to thank you again for your valuable feedback to improve our work. We are wondering whether our response has addressed your questions and can improve your opinion of our work.
>
> Kindly let us know if you might have further comments, and we will do our best to address them.
>
> Best Regards,
>
> The Authors

---

> > ### Author Response · Authors · 2024-12-01
> > **Kind Reminder for Discussion**
> >
> > Dear Reviewer DvW6,
> >
> > We have provided point-by-point responses to your concerns. We believe that our responses have addressed your concerns but still haven’t gotten any feedback from you. Do you have any further comments/suggestions?
> >
> > Best Regards,
> >
> > The Authors

---

> ### Author Response · Authors · 2024-12-03
> **Thank You for Raising Soundness Score & Your Valuable Feedback & Further Response - Part III**
>
> >**Q3**. ”I also have concerns regarding the potential impact of the Skeleton Semantics Learning (SSL) task on the broader machine learning community.”
>
> &
>
> >**Q4**. “Additionally, SSL presented in the paper is focused solely on human skeletons, the it appears to overlap with existing areas such as skeleton-based gait and action recognition.”
>
> **A3** & **A4**. Thank you for your constructive comments.
> - We would like to clarify that the proposed SSL task Prompter can be applied to not only skeleton-based person re-ID methods, but also gait recognition methods and action recognition methods. We have additionally evaluated the representative state-of-the-art **gait recognition** method GPGait [1] and **action recognition** method ST-GCN [2] on our different benchmark datasets, and integrated the proposed Prompter into them to verify its general applicability. As shown in the table below, the experimental results further demonstrate the effectiveness and generality of our method (i.e., the proposed spatial-temporal skeleton semantics learning) when applied to different architectures from varying research communities (e.g., state-of-the-art gait recognition method and representative action recognition method) to improve their performance (Rank-1 accuracy) in most cases under the same evaluation setting (highlighted in Appendix I, Sec. E.5).
>
> | Method Source (Research Community) | Method            | KS20 | IAS-A | IAS-B | KGBD |
> |------------------------------------|-------------------|:----:|:-----:|:-----:|:----:|
> |  Person Re-Identification          | TranSG [3]        | 71.3 |  48.0 |  56.1 | 57.0 |
> |                                    | + Prompter (Ours) | **74.2** |  **49.5** |  **60.4** | **59.5** |
> |  Gait Recognition                  | GPGait [1]        | 71.4 |  50.9 |  60.1 | 53.6 |
> |                                    | + Prompter (Ours) | **72.7** |  **55.3** |  **61.7** | 53.4 |
> |  Action Recognition                | ST-GCN [2]        | 60.4 |  43.6 |  49.1 | 57.7 |
> |                                    | + Prompter (Ours) | **65.6** |  **53.4** |  **58.8** | **59.0** |
>
> - With a more comprehensive evaluation of its effectiveness on methods from different areas (i.e., skeleton-based person re-ID, gait recognition, action recognition), we believe that the generality of the proposed SSL task Prompter is not limited to our main area, but it also can be widely applied to different architectures, related areas, and different machine-learning communities. We sincerely hope that the proposed first SSL generality assessment framework SCUT can inspire researchers to explore more useful SSL tasks and its broader application for different pattern recognition tasks, and wish that the devised general SSL task Prompter could advance the development of related machine-learning communities.
>
>
> >**Q5**. ” I believe the authors could further distinguish the SSL task by extending the analysis of skeletons to rigid, affine, or articulated objects, which might help demonstrate the unique contribution of SSL and its broader applicability.”
>
> **A5**. Thank you for your valuable and insightful suggestion! We will further extend the proposed SSL task from skeletons to more areas (e.g., rigid, affine, or articulated objects) in our future works. As mentioned in the section of “Broader Impacts” (please see Sec. D of Appendix D), the proposed Prompter can be potentially generalized to semantics learning of different fields (e.g., masked context reconstruction of 3D point clouds), and it can also serve as an effective representation-level augmentation strategy to combine with model-level augmentations such as Dropout algorithms to help reduce model over-fitting and improve their robustness against random perturbations. We promisingly envision its broader application in different areas and will proactively conduct such exploration in our next work.
>
> **We deeply thank the reviewer again for detailed comments and valuable advice to improve our work. We sincerely hope our clarifications above have addressed your concerns and can improve your opinion of our work.**
>
> References:
>
> [1] Yang et al. "Gpgait: Generalized pose-based gait recognition." ICCV, 2023.
>
> [2] Yan et al. "Spatial temporal graph convolutional networks for skeleton-based action recognition." AAAI, 2018.
>
> [3] Rao et al. "TranSG: Transformer-Based Skeleton Graph Prototype Contrastive Learning with Structure-Trajectory Prompted Reconstruction for Person Re-Identification." CVPR, 2023.

---

### Official Review · Reviewer_BHkC · 2024-11-04

**Soundness:** 3
**Presentation:** 3
**Contribution:** 3
**Rating:** 6
**Confidence:** 3

**Summary:**

In this paper, the authors propose the SCUT framework that identifies four key properties (STE, CTC, UT, TT) to assess the generality of skeleton semantics learning (SSL) tasks across different models and scenarios.

Based on SCUT, they further devise a generic SSL task, i.e., Prompter, to probabilistically and independently mask the spatial context of structural locations and temporal context of motion trajectories, which are exploited to reconstruct and infer complete skeleton sequences to capture general effective spatial-temporal skeleton semantics for person re-ID.

**Strengths:**

+ The insight of this work is novel and distinctive.

+ The writing of this paper is satisfactory.

+ The proposed SCUT is the first SSL generality assessment framework, which may be useful for the community.

+ The proposed Prompter is effective and adaptive generally for person Re-ID methods as shown in the experimental results.

**Weaknesses:**

- The proposed Co-Training Compatibility (CTC), Spatial-Temporal Effectiveness (STE), Task Transformability (TT), and Unsupervised Trainability (UT) are reasonable for SSL assessment. My question is whether they are necessary or complete for it.

- The proposed method Prompter is straightforward, which is a classical mask-and-reconstruct strategy.

- The Qualitative Attributes including UT and TT are also very simple to set as Yes or No.

- The different ranges and calculation methods of the 4 terms in SCUT make the parameter determination among these four terms is not easy.

- The paper layout can be further improved. For example, Tables 1 and 2 are far from their descriptions.

**Questions:**

As shown in Eqs. (4) and (5), we can see that G_ST has included G_C. This way, whether such two terms are both required?

---

> ### Author Response · Authors · 2024-11-25
>
> >**Q1**. "The proposed Co-Training Compatibility (CTC), Spatial-Temporal Effectiveness (STE), Task Transformability (TT), and Unsupervised Trainability (UT) are reasonable for SSL assessment. My question is whether they are necessary or complete for it."
>
> &
>
> >**Q2**. "The Qualitative Attributes including UT and TT are also very simple to set as Yes or No."
>
> **A1** & **A2**. Thank you for your valuable questions. We hope to highlight one of our key contributions is for the first time proposing a systematic generality assessment framework (SCUT) that identifies and quantifies key characteristics of SSL tasks. We systematically define two key properties (CTC and STE) that can be quantitatively computed to compare, and two qualitative characteristics (UT and TT) that can be evaluated in a unified and intuitive manner. It is worth mentioning that the task transformability (TT) may require a mathematical or theoretical analysis (please see Line 402-413 and Appendix II) to verify, so it is not a simple attribute for directly inferring, and we also provide detailed definitions and examples for researchers to identify this attribute (please see Line 238-245). All of these four criteria are necessary and can be combined to measure the final generality score of an SSL task in a unified formula (please see Eq. (6), Line 250-257, Table 1).
>
>
> >**Q3**. "The proposed method Prompter is straightforward, which is a classical mask-and-reconstruct strategy."
>
> **A3**. Thank you for your comments.
>
> - Firstly, compared with existing spatial-temporal masking strategies [1]-[2], we hope to clarify that the key novelty of the proposed Probabilistic Spatial Context Masking (PSCM) and Probabilistic Temporal Context Masking (PTCM) is that they are devised at an independent level of body structural locations and motion trajectory positions based on independent and identically distributed (IID) Bernoulli random masks. It possesses higher generality than previous methods (detailed in Sec. 3.2) and can be probabilistically generalized to different existing masking mechanisms for more effective skeleton semantics learning (please see TT property and Line 367-377). By contrast, existing masking strategies such as MS2L [2] directly masks the later consecutive skeletons (i.e., 150 frames) for temporal predictions while failing to learn effective spatial relations (i.e., performance degrades) under the used spatial masking. In [1], the structural positions are masked conditioned on the temporally-masked frames. Such direct frame-level or conditioned masking has several limitations, such as they cannot explicitly and individually model effective spatial semantics, nor can it feasibly evaluate the performance contribution of spatial masking (please refer to STE property defined in our work), while our method has solved these challenges with a focus of more generalizable skeleton context reconstruction.
>
> - Secondly, in our experiments, we also systematically compare our method with state-of-the-art SSL tasks using different masking strategies: Direct temporal masking (MIC), random masking with fixed-number masks (STPR), and the baseline without masking (DR). The experimental results demonstrate the higher effectiveness of our approach that adopts independent and finer-grained spatial-temporal masking. Moreover, we hope to highlight another novel contribution of our work is to propose a systematic SSL generality assessment framework (SCUT) to explore the multi-faceted performance and bottlenecks of existing SSL tasks under varying models and scenarios (please see Line 58-76). Motivated by the identified key properties of SCUT, we focus on devising a general solution (Prompter) that can be flexibly applied to different state-of-the-art skeleton-based person re-ID models (e.g., graph transformer, GAT, Siamese encoders). This is fundamentally different from MS2L [1] and [2] that rely on a certain action recognition backbone or model (i.e., GRU or STTFormer) to design effective masking strategies. Therefore, our method could be more general and scalable than these methods. Prompter can also be flexibly applied to RGB-estimated skeletons, unsupervised scenarios, different graph modeling, and cross-domain person re-ID tasks (see Further Analyses in our paper).

---

> ### Author Response · Authors · 2024-11-25
>
> >**Q4** ”The different ranges and calculation methods of the 4 terms in SCUT make the parameter determination among these four terms is not easy.”
>
> **A4**. Thank you for your valuable feedback. For the first two quantitative properties, the performance gains can be obtained based on the evaluation results on different datasets, while the weight coefficient is typically set to 1 as we can consider each applied base model and dataset equally significant. In this sense, there are no extra parameters that require the users to determine for computing G_C (i.e., Co-training compatibility (CTC)) (please see Eq. (3)). In G_ST (i.e., spatial-temporal effectiveness (STE)), it shares identical parts (e.g., average performance gain) that can be re-used from G_C, while it only requires users to compute one more key factor R^{ST}_{i, j} (please see Eq. (5)), which can also be directly calculated based on the experimental results evaluated on two parts (spatial part and temporal part) of SSL. Both G_C and G_ST have the same value range in (-1, 1) as we have integrated a value normalization into the corresponding computing formula. For another two qualitative attributes, they have no extra parameters (e.g., performance gain) to determine or compute. Based on the above reasons, we believe that the defined four terms in SCUT are relatively easy to determine compared with other performance or feature-based metrics that require extra operations (e.g., query and match) on all feature representations.
>
> >**Q5**. "The paper layout can be further improved. For example, Tables 1 and 2 are far from their descriptions."
>
> **A5**. Thank you for your valuable suggestion. We have revised the paper layout to put Tables 1 and 2 to be closer to their descriptions (highlighted in blue) while maximizing the utilization efficiency of paper space.
>
>
> >**Q6**. "As shown in Eqs. (4) and (5), we can see that G_ST has included G_C. This way, whether such two terms are both required?"
>
> **A6**. Thank you for your valuable question. We would like to clarify that the common part (i.e., the last fraction formula) in both G_ST and G_C is to compute the average performance improvement after applying the SSL task. However, in G_C, we focus on the average co-training performance gain under different models and datasets, so we incorporate the respective gain of a certain model on a specific dataset with corresponding weight coefficient (please see Eq. (3), Line 181-189, Sec. 3.2), so as to measure whether and how much performance gain in average that an SSL task can compatibly facilitate on different combinations of models and datasets. Different from G_C that computes the overall performance gain, the proposed spatial-temporal effectiveness (STE), i.e., G_ST, aims to quantify the average spatial-temporal performance gain. It contains a unique component R^{ST}_{i, j} to compute the relative ratio between the performance gains of spatial part and temporal part of an SSL task (please see Eq. (5), Line 209-230, Sec. 3.2). This component not only considers independent performance gain from spatial or temporal component (e.g., spatial/temporal semantics learning) of an SSL task, but also synergizes their contribution balance to encourage a more reasonable measurement of spatial-temporal effectiveness (please see definition and motivation in Line 213-233). Therefore, G_ST is different from G_C in terms of motivation, definition, and the key part of computation. To simultaneously evaluate the overall co-training performance gain as an indicator of SSL compatibility (CTC), and measure the combined effectiveness of independent spatial and temporal parts of SSL to indicate the spatial-temporal modeling effectiveness (STE) of SSL, both G_ST and G_C are necessary in the proposed SSL generality assessment framework.
>
>
> References:
>
> [1] Wu, Wenhan, et al. "Skeletonmae: Spatial-temporal masked autoencoders for self-supervised skeleton action recognition." 2023 IEEE international conference on multimedia and expo workshops (ICMEW), IEEE, 2023.
>
> [2]Lin, Lilang, et al. "Ms2l: Multi-task self-supervised learning for skeleton based action recognition." Proceedings of the 28th ACM international conference on multimedia, 2020.

---

> > ### Comment · Reviewer_BHkC · 2024-11-26
> > **Comments after rebuttal**
> >
> > Thanks for the authors' response, which has partly addressed my concerns.
> >
> > As a new SSL assessment framework, the proposed four metrics are reasonable, although not very mature and objective enough.
> >
> > I acknowledge the specific differences between the proposed method and [1,2], for the SSL problem. However, I can not find enough technical novelty that could be common for the machine-learning community, since its rationale (spatial-temporal mask-and-reconstruct strategy) is not very novel. It seems that the application of this strategy makes the proposed Prompter more flexible.
> >
> > Overall consideration, I'm sorry, but I can not give a higher score for this work.

---

> ### Author Response · Authors · 2024-11-28
> **Response A: Thank You for Your Valuable Feedback & Further Response - Part I**
>
> >”Thanks for the authors' response, which has partly addressed my concerns.“
>
> We would like to thank you again for your valuable feedback to improve our work. As we have provided point-by-point responses to your concerns, please kindly let us know if you might have further comments, and we will do our best to address them.
>
>
> >**Q1**. ”However, I can not find enough technical novelty that could be common for the machine-learning community, since its rationale (spatial-temporal mask-and-reconstruct strategy) is not very novel. It seems that the application of this strategy makes the proposed Prompter more flexible.”
>
> **A1**. Thank you for your valuable feedback and for recognizing the flexibility of our method. We hope to clarify that apart from the key differences (i.e., independent masking of body structural locations and motion trajectory positions based on independent and identically distributed (IID) Bernoulli random masks) from existing masking mechanisms (detailed in the previous response), the main novelty of Prompter include:
> - **Explicit** effective temporal and spatial modeling (i.e., implement spatial-temporal effectvienss (STE)) in terms of body structure and motion trajectory (we have verfiies the effectiveness of each part in Abalation study), unlike previous SSL tasks used in skeleton-based person re-ID that do not distinguish these two parts for reconstruction (see Sec. 3.3);
>  - Fully exploit varying valuable **context** information (e.g., temporal context of trajectory) of **fine-grained** skeleton representations to capture richer skeleton semantics (please see Line 299-309), which is achieved by combining multiple skeleton context based learning sub-objectives (i.e., establish Task Transformability (TT)), while existing SSL tasks typically utilize a fixed reconstruction objective (e.g., direct reconstruction or with fixed masks).
> - Moreover, it is also a general SSL task that does not rely on any specific model architectures or feature representations, which is inspired and designed by the crucial properties/principles of SSL identified by SCUT (please see Line 302-309). This also suggests the potential value of the proposed SCUT framework to devise more general SSL tasks for different scenarios. It can also be potentially modeled as a **novel model regularization method** like Dropout (please see assumptions and analyses in Appendix II, and we will provide further discussion in the future version).

---

> ### Author Response · Authors · 2024-11-28
> **Response A: Thank You for Your Valuable Feedback & Further Response - Part II**
>
> - Apart from the above novelty, we would like to highlight the **generality** (which is also the key focus of our study) of the proposed SSL task Prompter for different machine-learning communities, which can be applied to not only skeleton-based person re-ID methods, but also gait recognition methods and action recognition methods. We have additionally evaluated the representative state-of-the-art **gait recognition** method GPGait [1] and **action recognition** method ST-GCN [2] on different benchmark datasets, and integrated the proposed Prompter into them to verify its general applicability.
>
> As shown in the table below, the experimental results further demonstrate the effectiveness and generality of our method (i.e., the proposed spatial-temporal skeleton semantics learning) when applied to different architectures from varying research communities (e.g., state-of-the-art gait recognition method and representative action recognition method) to improve their performance (Rank-1 accuracy) in most cases under the same evaluation setting (highlighted in Appendix I, Sec. E.5).
>
> | Method Source (Research Community) | Method            | KS20 | IAS-A | IAS-B | KGBD |
> |------------------------------------|-------------------|:----:|:-----:|:-----:|:----:|
> |  Person Re-Identification          | TranSG [3]        | 71.3 |  48.0 |  56.1 | 57.0 |
> |                                    | + Prompter (Ours) | **74.2** |  **49.5** |  **60.4** | **59.5** |
> |  Gait Recognition                  | GPGait [1]        | 71.4 |  50.9 |  60.1 | 53.6 |
> |                                    | + Prompter (Ours) | **72.7** |  **55.3** |  **61.7** | 53.4 |
> |  Action Recognition                | ST-GCN [2]        | 60.4 |  43.6 |  49.1 | 57.7 |
> |                                    | + Prompter (Ours) | **65.6** |  **53.4** |  **58.8** | **59.0** |
>
> With a more comprehensive evaluation of its effectiveness on methods from different areas (i.e., skeleton-based person re-ID, gait recognition, action recognition), we believe that the flexibility of the propsoed SSL task Prompter is not limited to our main area, but it also can be widely applied to different architectures, related areas, and different machine-learning communities. We sincerely hope that the proposed first SSL generality assessment framework SCUT can inspire researchers to explore more useful SSL tasks and its broader application for different pattern recognition tasks, and wish that the devised general SSL task Prompter could advance the development of related machine-learning communities.
>
> We thank the reviewer again for the detailed comments and valuable questions. We sincerely hope our clarifications above have addressed your concerns and can improve your opinion of our work.
>
> References:
>
> [1]Yang et al. Gpgait: Generalized pose-based gait recognition. ICCV, 2023.
>
> [2]Yan et al. Spatial temporal graph convolutional networks for skeleton-based action recognition. AAAI, 2018.
>
> [3] Rao et al. TranSG: Transformer-Based Skeleton Graph Prototype Contrastive Learning with Structure-Trajectory Prompted Reconstruction for Person Re-Identification. CVPR, 2023.

---

> ### Comment · Reviewer_BHkC · 2024-11-28
> **Final evaluation**
>
> First, I have no other new comments but **response does not mean address**.
>
> Next, about the novelty of the proposed method, I am still not persuaded.
>
> I do not deny the effectiveness of the proposed method for temporal and spatial modeling and skeleton representations. However, **effective does not equal novel/original**. Once again, the insight of spatial-temporal mask-and-reconstruct strategy is not unexpected for the SSL.
>
> The authors take the Dropout for example, which may over-claim the contribution. Dropout has been verified in lots of deep-learning models, but the proposed method is only used for SSL-based human-related tasks, which is not general enough as Dropout. But I also approve of the generality of the Prompter for different downstream tasks provided by the authors' response part II.
>
> I provide a rating of 6, which can not be improved considering the overall contributions and novelty.
>
> Thanks.

---

> ### Author Response · Authors · 2024-11-28
> **Response B: Thank You for Your Valuable Feedback & Further Response**
>
> Thank you for your valuable feedback. We would like to provide a summary of both novelty and generality of our method for reviewers and area chair:
>
> ### Novelty:
> 1. Our work is **the first exploration** and assessment of multi-faceted generality of existing skeleton semantics learning (SSL) tasks under different scenarios, and presents the **first systematic generality assessment framework** termed SCUT that identifies and quantifies key characteristics of SSL tasks in terms of Spatial-temporal effectiveness (STE), Co-training compatibility (CTC), Unsupervised trainability (UT), and Task transformability (TT). Motivated by SCUT, we propose a generic Probabilistic Masked Spatial-Temporal Context Reconstruction (Prompter) task to enhance general skeleton semantics learning of different models for person re-ID.
>
>  2. Unlike existing masking-reconstruction mechanisms that directly mask a portion of consecutive frames [1], employ a fixed number of masks [2,3] or non-independent sequential context (e.g., fixed subsequences) for masking [4], the proposed SSL Task Prompter performs skeleton context masking at the **independent joint level** based on independent and identically distributed (IID) Bernoulli random masks (please see Line 318-323, Eq. (7), Line 332-334. Eq. (8)), which is also **the first exploration** of independent probabilistic masking mechanism in the area of skeleton-based person re-ID.
>
> 3. Spatial-Temporal Effectiveness (STE) (defined in the explored SCUT framework): The proposed Prompter explicitly models temporal and spatial semantics **separately** and **jointly** from body structure (i.e., spatial structural locations, please see Line 310-313) and motion trajectory (i.e., temporal joint positions, please see Line 324-326, 352-366).
>
> 4. Task Transformability (TT) (defined in the explored SCUT framework): Prompter fully exploits **varying valuable context** information (e.g., random temporal context of trajectory) of **fine-grained** skeleton representations to capture richer skeleton semantics (please see Line 299-309, Line 367-377 of our paper), which is achieved by combining multiple skeleton context based learning sub-objectives, while existing SSL tasks typically utilize a single or fixed reconstruction objective (e.g., direct reconstruction or with fixed masks).
>
> ### Generality:
> 1. The proposed SSL generality assessment framework SCUT can be applied to evaluate **multi-faceted generality** of different SSL tasks used for skeleton-based person re-ID and other areas.
>
> 2. The proposed SSL task Prompter possesses **higher overall generality scores** than existing SSL tasks in four aspects of Spatial-temporal effectiveness (STE) (i.e., more than twice the performance gain than the latest SSL task STPR), Co-training compatibility (CTC) (i.e., up to 6.84% average performance gain on varying models and datasets), Unsupervised trainability (UT), and Task transformability (TT) (please see Line 261-291, Table 1).
>
> 3. The proposed Prompter can be probabilistically **generalized** to different existing masking-based SSL tasks (defined as sub-tasks), and we show that it can enable the model to jointly learn with contained sub-tasks (please see Line 367-377, Line 524-528, Fig. 3).
>
> 4. Prompter can be widely applied to different types of methods, including **unsupervised** methods (e.g., SPC-MGR, SimMC), **self-supervised** and **supervised** methods (e.g., MG-SCR, TranSG).
>
> 5. Prompter can be applied to **different skeleton modeling** (i.e., joint-level, part-level, body-level human body representation), as shown in Line 519-523, Fig. 2 and Appendix I Table 11.
>
> 6. Prompter can be applied to both sensor-based skeleton data (main experiments) and **RGB-estimated skeleton data** (please see Line 505-512, Table 5).
>
> 7. Prompter can be applied to **cross-domain person re-ID** tasks to learn general effective skeleton semantics (please see Line 513-518, Table 6 and Appendix I Fig. 11, 12, 13).
>
> 8. Prompter can be potentially modeled as a general **model regularization** method (please see Line 374-377), and we further reveal its relations to existing regularization methods (e.g., Dropout) in Appendix II.
>
> 9. Prompter can be generally applied to different methods/architectures from **varying research communities** (e.g., skeleton-based person re-ID, gait recognition, action recognition) to improve their performance (please see the previous response and Appendix I, Sec. E.5, Table 13, 14). It could promisingly be applied to other tasks and advance the development of related research.
>
> References:
>
> [1] Wu et al. "SkeletonMAE", ICMEW, 2023.
>
> [2] Lin et al. "MS2L", ACMMM, 2020.
>
> [3] Rao et al. “SimMC”, IJCAI, 2022.
>
> [4] Rao et al. “TranSG”, CVPR, 2023.
>
> **We deeply appreciate the reviewer's detailed feedback and valuable comments. We sincerely hope reviewers and the area chair can take our above clarifications into consideration.**
>
> Best Regards,
>
> The Authors

---

> > ### Comment · Reviewer_BHkC · 2024-12-02
> > **Conclusive Review**
> >
> > First, the reviewer thanks the authors' unremitting efforts.
> >
> > About novelty, any work has its novelty (even the first exploration) as long as the involved research area is fine enough.
> >
> > About the generality, the applications of one technology to different related downstream tasks are helpful. But the insights that can inspire more researchers in the machine-learning community are more significant. The research on SSL is somewhat narrow, in which the related works (last paragraph in Sec. 2) and comparison methods (Tables 2,3, and 5) are all from the same research group, even the same first author (also pointed out by other reviewers). So this work may contribute to SSL, but how much to contribute to the ML/CV/AI community is uncertain.
> >
> > Again, I will not change (may increase or decrease) my final rating, and I don't want to impact the evaluation of other reviewers.
> >
> > This is my final comment, please leave it in the end, with **No** reply required. The overall decision is left to AC.
> >
> > Thanks.

---

> ### Author Response · Authors · 2024-12-02
> **Response C: Thank You for Your Valuable Feedback & Further Response - Part I**
>
> Dear Reviewer,
>
> We are heartfelt grateful for your detailed feedback and would like to thank you again for appreciating our efforts. We hope to answer your further questions below.
>
> >**Q1**. ”the related works (last paragraph in Sec. 2) and comparison methods (Tables 2,3, and 5) are all from the same research group, even the same first author (also pointed out by other reviewers). So this work may contribute to SSL, but how much to contribute to the ML/CV/AI community is uncertain.”
>
> **A1**. Thank you for your valuable comment.
> - Firstly, we would like to point out that **our method is compared with all existing state-of-the-art skeleton-based person re-ID methods, regardless of whether they belong to the same or different groups** (i.e., methods from the same or different groups will not influence our selection and comparison criteria based on the sufficient technical soundness, novelty, and impact of these methods). All methods are fairly compared using the same evaluation protocol.
>
> - Secondly, **our method is actually compared with 20 approaches from diverse groups of authors (up to 13 different groups), instead of from the same research group**, and here we provide an overview of compared methods (we present the most representative method from the same group and also include newly added during the revision) in our work for both reviewers and area chair:
>
> | Comparison Part                                    | Method  Type                          | Method       | Source Group                          |
> |----------------------------------------------------|---------------------------------------|--------------|---------------------------------------|
> | Sec. 4.2 & Revision (Added to Appendices or Paper) | Skeleton-Based Person Re-ID Method    | D_PG         | Liao et al., 2020 (Totally 2 Methods) |
> |                                                    |                                       | D_13         | Munaro et al., 2014                   |
> |                                                    |                                       | D_16         | Pala et al., 2019                     |
> |                                                    |                                       | TranSG       | Rao et al., 2023 (Totally 7 Methods)  |
> |                                                    | Skeleton-Based Gait Recognition Method | SkeletonGait | Chao et al., 2024                     |
> |                                                    |                                       | GaitTR       | Cun et al., 2023                      |
> |                                                    |                                       | GPGait       | Yang et al., 2023                     |
> |                                                    | Action Recognition Method             | ST-GCN       | Yan et al., 2018                      |
> | Sec. 4.3                                           | Person Re-ID Method                   | LMNN         | Weinberger & Saul, 2009               |
> |                                                    |                                       | ITML         | Davis et al., 2007                    |
> |                                                    |                                       | ELF          | Gray & Tao, 2008                      |
> |                                                    |                                       | SDALF        | Farenzena et al., 2010                |
> |                                                    |                                       | MLR          | Liu et al., 2015                      |
>
> - **We acknowledge that seven state-of-the-art skeleton-based person re-ID methods (*35% of all methods*) originate from the same group**, which developed **the first skeleton-based person re-ID method as early as 2020 [1]** and has pioneered numerous skeleton-based innovations in both person re-ID and action recognition communities [2]. It is worth noting that most existing skeleton-based person re-ID works also follow the settings and designs from this research group. However, with a systematic evaluation of the skeleton semantics learning (SSL) tasks used in these methods based on the proposed SCUT framework, we **identify the bottlenecks in their generality and performance**, including the lack and limit of co-training compatibility (CTC) and spatial-temporal effectiveness (STE) (please see Line 70-76, Table 1 and Line 367-377 of our paper), which might negatively influence their performance when applied to different models and scenarios (e.g., different types of skeleton data) (please see Table 3, 5 and Sec. 4.3 of our paper). This motivates us to propose the general SSL task Prompter to address these challenges to facilitate the performance of different state-of-the-art skeleton-based person re-ID models under different scenarios (please see Line Sec. 4.2, Table 2 of our paper).

---

> ### Author Response · Authors · 2024-12-02
> **Response C: Thank You for Your Valuable Feedback & Further Response - Part II**
>
> - Moreover, **our work is also the first exploration and assessment of the multi-faceted generality of existing SSL tasks under different scenarios, and presents the first systematic generality assessment framework (SCUT)** that identifies and quantifies key characteristics of SSL tasks. In our paper and its revised version, we have not only **(1) compared with the methods from three different areas such as person re-ID, gait recognition, action recognition** (see the above table) to demonstrate the superiority of our method, but also **(2) integrated the proposed SSL task Prompter into methods from these three different areas** to verify the effectiveness of our method in improving their performance (please see response to Reviewer DvW6, iRXh, DUn6). Therefore, we believe that both the proposed SSL generality assessment framework and the proposed Prompter method could provide valuable insights and have the potential to advance broader AI/CV related areas.
>
> References:
>
> [1] Rao et al. “Self-supervised gait encoding with locality-aware attention for person re-identification.” IJCAI, 2020.
>
> [2] Rao et al. “Augmented Skeleton Based Contrastive Action Learning with Momentum LSTM for Unsupervised Action Recognition” Information Sciences, 2021.
>
> **We deeply appreciate the reviewer's detailed feedback and valuable comments to improve our work**. In our future research on SSL, we will include more up-to-date methods from various related areas (e.g., gait recognition, pose estimation) for a more comprehensive evaluation and comparison, to benefit and facilitate a larger ML/CV/AL community.
>
> We sincerely hope reviewers and the area chair (AC) can take our above clarifications into consideration, and we believe that AC will make a fair decision.
>
> Best Regards,
>
> The Authors

---

> ### Comment · Reviewer_BHkC · 2024-12-02
> **No further discussion required**
>
> The reviewer hopes the authors no longer reply if they have read the comments carefully.
>
> About novelty, any work has its novelty (even the first exploration) as long as the involved research area is fine enough.  I can not find enough technical novelty that could be common for the machine-learning community since its rationale (spatial-temporal mask-and-reconstruct strategy) is not very novel. It seems that the application of this strategy makes the proposed Prompter more flexible.
>
> About the generality, the applications of one technology to different related downstream tasks are helpful. But the insights that can inspire more researchers in the machine-learning community are more significant. The research on SSL is somewhat narrow, in which the related works (last paragraph in Sec. 2) and comparison methods (Tables 2,3, and 5) are all from the same research group, even the same first author (also pointed out by other reviewers). So this work may contribute to SSL, but how much to contribute to the ML/CV/AI community is uncertain.
>
> Again, I don't want to change my final rating and impact the evaluation of other reviewers.
>
> This is my final comment. **Please leave it in the end, with No further reply required**. The overall decision is left to AC.
>
> Thanks again.

---

### Author Response · Authors · 2024-12-03
**General Response**

Dear ACs and Reviewers,

We sincerely appreciate your time and effort in reviewing our paper and providing constructive feedback. Besides the response to each reviewer, here we would like to further 1) thank reviewers for their recognition of our work and 2) highlight the major modifications in our revision:

1. **We are glad that the reviewers appreciate and recognize our novelty and contributions**.

    - The idea and insight of this work is **novel/new**, **distinctive**, and **exciting**, such as the assessment framework aspect. [v2zj, BHkC, iRXh]

    - The proposed SSL generality assessment framework can/may be **helpful** in the community such as skeleton representation learning field, and can be applied to **many other tasks** as well. [DUn6, v2zj, BHkC]

    - The unified framework has the potential to **advance the development of this field**, not only in the skeleton-based person re-identification area but also in the **whole skeleton-based research community**. [v2zj]

    - The proposed Probabilistic Masked Spatial-Temporal Context Reconstruction (Prompter) task is **technically reasonable**, **effective/robust** [BHkC, DvW6], and **geneirc** with a **great improvement** when applied to different existing methods. [DUn6, v2zj]

    - The proposed Prompter method has shown **encouraging results** on methods from **different areas** such as gait recognition and action recognition [iRXh], and demonstrated potential **generality** for different downstream tasks. [BHkC]

    - This paper is well-written and easy to follow. [DvW6, DUn6, BHkC]

    - The experiments are **extensive** and **thorough**, providing **convincing** evidence of the proposed method's **effectiveness/robustness** on different model architectures and benchmakrs arcoss various settings. [DvW6, iRXh, DUn6]


2. **We summarize the main modifications in our revised paper (highlighted in blue)**.

    - We add a comparison of key differences and similarities between our method (i.e., skeleton-based person re-ID) and skeleton-based gait recognition methods in Appendx I Sec. E.1 to help readers better understand these two areas. [iRXh]

    - We add more results of different state-of-the-art gait recognition methods (SkeletonGait, GaitTR, GPGait) on KS20, BIWI, IAS, KGBD datasets, and compare them with our method in Appendx I Sec. E.1. The new results further verify the effectiveness and efficiency of our proposed method. [iRXh]

    - We add more performance results of different SSL tasks (DR, MIC, STPR, Prompter) under different skeleton levels (Joint-Level, Part-Level, Body-Level) on different datasets in Appendx I Sec. E.2, which further verifies the effectiveness and necessity of joint-level reconstruction in both the proposed method and other SSL tasks. [DvW6]

    - We add more qualitative examples and analyses for the cross-domain person re-ID performance, including confusion matrices and t-SNE feature visualization, in Appendx I Sec. E.2. The new results are consistent with the performance results in Table 6 and also provide valuable insights into the learned features and generalized performance across datasets. [DvW6]

    - We add an overview of state-of-the-art skeleton semantics learning (SSL) tasks, their source method, and method types in Appendx I Sec. E.3, to provide a clearer method classification. [v2zj]

    - We add more discussions regarding the differences between our method and existing state-of-the-art masking strategies in Appendx I Sec. E.3.1, elaborating on the novelty and generality of our method. [iRXh, BHkC, v2zj]

    - We add new results of evaluating the state-of-th-art and representative gait recognition method such as GPGait and action recognition method such as ST-GCN on different datasets, and add new results of integrating the proposed Prompter into these methods in Appendx I Sec. E.4 and E.5. The new results further verify the effectiveness and **generality of our method to improve the gait recognition and action recognition method**. [DUn6, iRXh, DvW6]

    - We revise the paper content such as related works in Sec. 2, and paper layout such as table positions in Sec. 4.2, to make it more informative and readable. [DUn6,BHkC]


Best regards,

The Authors

---

### Meta-Review · Area_Chair_ztfp · 2024-12-19

**Metareview:**

This paper introduces a framework for evaluating the generality of spatial-temporal semantics learning (SSL) tasks in skeleton-based person re-identification and proposes Prompter, a probabilistic masked spatial-temporal context reconstruction task to enhance model performance by capturing fine-grained skeleton patterns. Empirical results demonstrate its superiority across diverse datasets and scenarios.

The paper is well-structured, with clear and easy-to-follow writing. The experiments are extensive , providing strong evidence of the proposed method's effectiveness. The algorithm consistently outperforms baseline methods across diverse settings, showcasing its potential and robustness in skeleton-based person re-identification.

The proposed generality assessment metrics for skeleton semantic learning, while reasonable, are neither fully mature nor sufficiently objective. Moreover, the Prompter method is a classical mask-and-reconstruct strategy, which limits its novelty and overall contribution to the ML/CV/AI community. Additionally, skeleton-based person re-identification is fundamentally similar to skeleton-based gait recognition. Separating these two closely related research directions and developing distinct evaluation systems for each is argued to be unproductive for advancing progress in the field.

**Additional Comments On Reviewer Discussion:**

After rebuttal, several concerns raised by three reviewers remain unresolved:  1) Reviewers BHkC and DvW6 acknowledged the effectiveness of the proposed Prompter method for temporal and spatial modeling and skeleton representation. However, they noted that it employs a classical mask-and-reconstruct strategy, offering limited novelty and contribution to the ML/CV/AI community. 2) Reviewer iRXh expressed that skeleton-based person re-ID is fundamentally similar to skeleton-based gait recognition. They argued that separating these two research directions and developing distinct evaluation systems for each is not advantageous for advancing related research.

While two reviewers, v2zj and DUn6, recommended acceptance, the Area Chair (AC) concurred with Reviewers BHkC and DvW6 on the work's limited novelty. The AC also agreed with Reviewer iRXh regarding the overlap between skeleton-based person re-ID and gait recognition.

---

### Decision · Program_Chairs · 2025-01-22

Reject